# Integrated rupture mechanics for slow slip events and earthquakes

Huihui Weng [1,2] ✉ & Jean-Paul Ampuero [1]

Slow slip events occur worldwide and could trigger devastating earthquakes, yet it is still debated whether their moment-duration scaling is linear or cubic and a fundamental model unifying slow and fast earthquakes is still lacking. Here, we show that the rupture propagation of simulated slow and fast earthquakes can be predicted by a newly-developed three-dimensional theory of dynamic fracture mechanics accounting for finite rupture width, an essential ingredient missing in previous theories. The complete spectrum of rupture speeds is controlled by the ratio of fracture energy to energy release rate. Shear stress heterogeneity can produce a cubic scaling on a single fault while effective normal stress variability produces a linear scaling on a population of faults, which reconciles the debated scaling relations. This model provides a new framework to explain how slow slip might lead to earthquakes and opens new avenues for seismic hazard assessment integrating seismological, laboratory and theoretical developments.

Slow slip events (SSEs) have been observed in subduction megathrusts and crustal faults worldwide[1–9] and may trigger large megathrust earthquakes[10–13], therefore understanding the physical mechanisms of SSEs is of increasing importance. SSEs usually occur in an elongated section of the deep plate interface and have rupture speeds much slower than large megathrust earthquakes, whose ruptures are also elongated (Fig. 1a). A compilation of rupture speeds of global SSEs[14–16], earthquakes[17–21] and laboratory experiments[22] illustrates that spontaneous ruptures span a wide range of speeds, from ultra-slow speeds up to the P-wave speed (Fig. 1b). Previous simulations show that SSE ruptures can propagate steadily at a very slow speed if facilitated by a frictional transition from rate-weakening at low slip rates to rate-strengthening at high slip rates[23–26] or by fault gouge dilatancy with an associated change in fluid pressure[27–30], both of which are observed experimentally[30–41]. In addition, earthquake ruptures on long faults can steadily propagate at supershear speeds (faster than S-wave speeds), depending on the balance between fracture energy and energy release rate[42]. Though laboratory experiments[22,34,43,44] have suggested a continuum of rupture speeds, these experiments lack a finite rupture width, an essential ingredient of large slow and fast ruptures on natural faults, and the general rupture mechanics controlled by such finite rupture width is not completely understood. Empirical moment-duration scaling relations[14,45–52] have been used to compare the physical mechanisms of SSEs and earthquakes, yet it is still debated whether the scaling of SSEs is linear[45,46] or cubic[14,47,48] and a fundamental model that integrates SSEs and earthquakes is still lacking.

Here, we show that the rupture propagation of SSEs and earthquakes on long faults, with elongated ruptures as widely observed (e.g., refs. 14, 46, 53), can be predicted by the same theoretical equation of motion of the rupture tip and the debated scaling behaviours of SSEs can be attributed to different types of fault heterogeneities.

## Results

### General mechanics for steady SSEs and earthquakes

Numerical and theoretical studies[53,54] demonstrated that the energy balance governing rupture propagation on long faults with finite width in a 3D elastic medium depends on the rupture width rather than on the rupture length. In particular, Weng and Ampuero[53] developed a 3D theory of large ruptures that accounts for their finite rupture width $W$. Their theory yields a rupture tip equation of motion of the following form (Methods):

$$F\left(\frac{G_c}{G_0}\right) = M(v_r) \cdot \dot{v}_r, \tag{1}$$

[1]Université Côte d'Azur, IRD, CNRS, Observatoire de la Côte d'Azur, Géoazur, Valbonne, France. [2]School of Earth Sciences and Engineering, Nanjing University, Nanjing, China. ✉e-mail: weng@nju.edu.cn

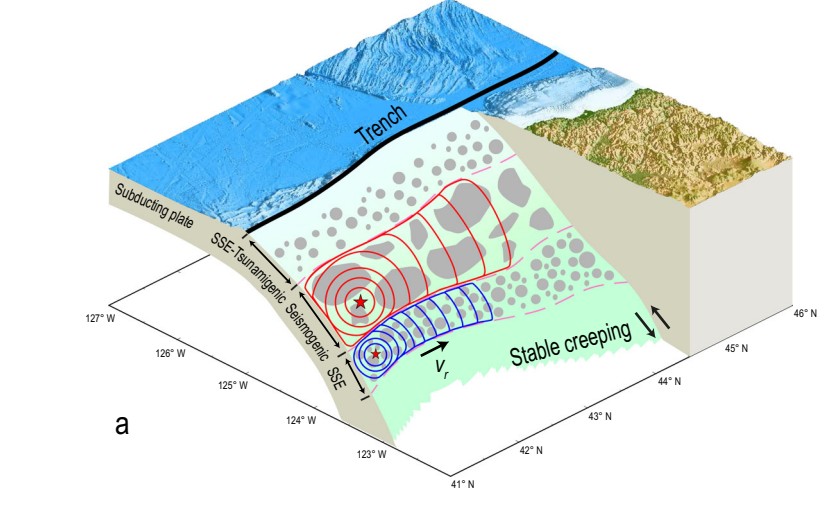

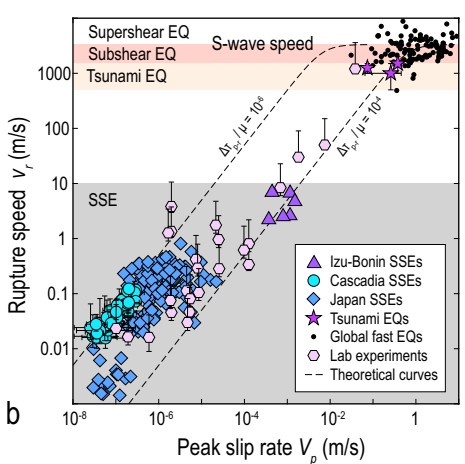

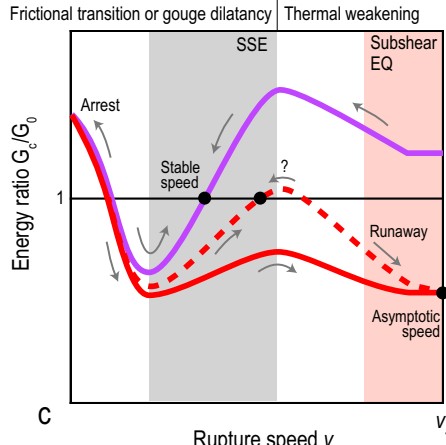

**Fig. 1 | Slow and fast ruptures on a subduction zone and their rupture speeds.**
**a** Sketch of subduction zone composed of tsunamigenic, seismogenic and SSE zones with finite widths. Rupture propagation fronts of SSE (blue curves) and earthquake (red curves) start at the hypocenters indicated by red stars. Topography data is downloaded from the NOAA Data Catalog. **b** Symbols represent estimates of rupture speed versus peak slip rate of observed SSEs[14–16], laboratory experiments[22], tsunami earthquakes[17–19] and regular earthquakes[20,21] (error bar indicates uncertainty when available; Methods). Dashed curves are theoretical predictions (Methods). $\Delta\tau_{p-r}$ and $\mu$ are the peak-to-residual strength drop at the rupture front and shear modulus, respectively. **c** Models for energy ratio $G_c/G_0$ integrating multiple frictional mechanisms. Purple curve represents one example of $G_c/G_0$ for SSEs. Red dash and solid curves represent examples of $G_c/G_0$ for earthquakes. Grey arrows indicate the evolution of rupture speed controlled by equation (1).

where $G_c/G_0$ is the ratio between the fracture energy and the energy release rate of steady-state ruptures, $\dot{v}_r$ is the rupture acceleration, the time derivative of rupture speed $v_r$, and $F$ and $M$ are known universal functions. Here, the energy release rate of steady-state ruptures is exactly equal to the static energy release rate and, in contrast with the classical 2D fracture mechanics theory, it is independent of the rupture length[53]: $G_0 = \Delta\tau^2 W/(\pi\mu)$, where $\Delta\tau$ is the static stress drop and $\mu$ is the shear modulus. These features of $G_0$ arise because the finite fault width $W$ turns the rupture into a slip pulse with length smaller than $W$, and are similar to those of a rupture in a 2D strip of thickness $W$ (refs. 55, 56). The finite $W$, an essential ingredient for the study of large earthquakes and SSEs, is missing in previous experimental and theoretical studies of slow and fast ruptures[43,44].

In general, both $G_c$ and $G_0$ can be functions of rupture speed $v_r$ and depend on the friction behaviour of the fault[57]. In particular, under certain rate-dependent friction laws, $G_0$ depends on $v_r$ via the dependence of static stress drop $\Delta\tau$ on slip rate, thus on $v_r$. Especially for a steady-state rupture, substituting $\dot{v}_r = 0$ into equation (1) yields the energy balance condition: $G_c(v_r) = G_0(v_r)$. Weng and Ampuero[53] proposed that steady-state ruptures can propagate at any speed up to the S-wave speed if the fracture energy increases with increasing rupture

speed. Further linear stability analysis (Methods) shows that the general stability condition for steady-state ruptures can be written as

$$\frac{d(G_c/G_0)}{dv_r} > 0. \qquad (2)$$

The opposite condition, $d(G_c/G_0)/dv_r \leq 0$, predicts non-steady ruptures that either accelerate ($G_c/G_0 < 1$) or decelerate ($G_c/G_0 > 1$). Generally, equation (1) predicts that there are two types of ruptures on long faults with finite width: steady and non-steady ruptures. The predicted features of non-steady ruptures with speeds typical of regular earthquakes (close to S-wave speed) have been numerically validated in a previous study[53], whereas the predicted steady-state ruptures of both fast and slow speeds are validated by numerical simulations here.

Here, we further validate equation (1) for a continuum of steady rupture speeds from arbitrarily-slow speeds up to the S-wave speed in numerical simulations (Fig. S1a). This model considers a rate-and-state friction law with rate-weakening behaviour at low slip rates and rate-strengthening behaviour above a critical slip rate $V_c$ (Methods), as observed in laboratory experiments[31–41]. We find that the finite-width fault, controlled by a rate-dependent friction, also turns the dynamic

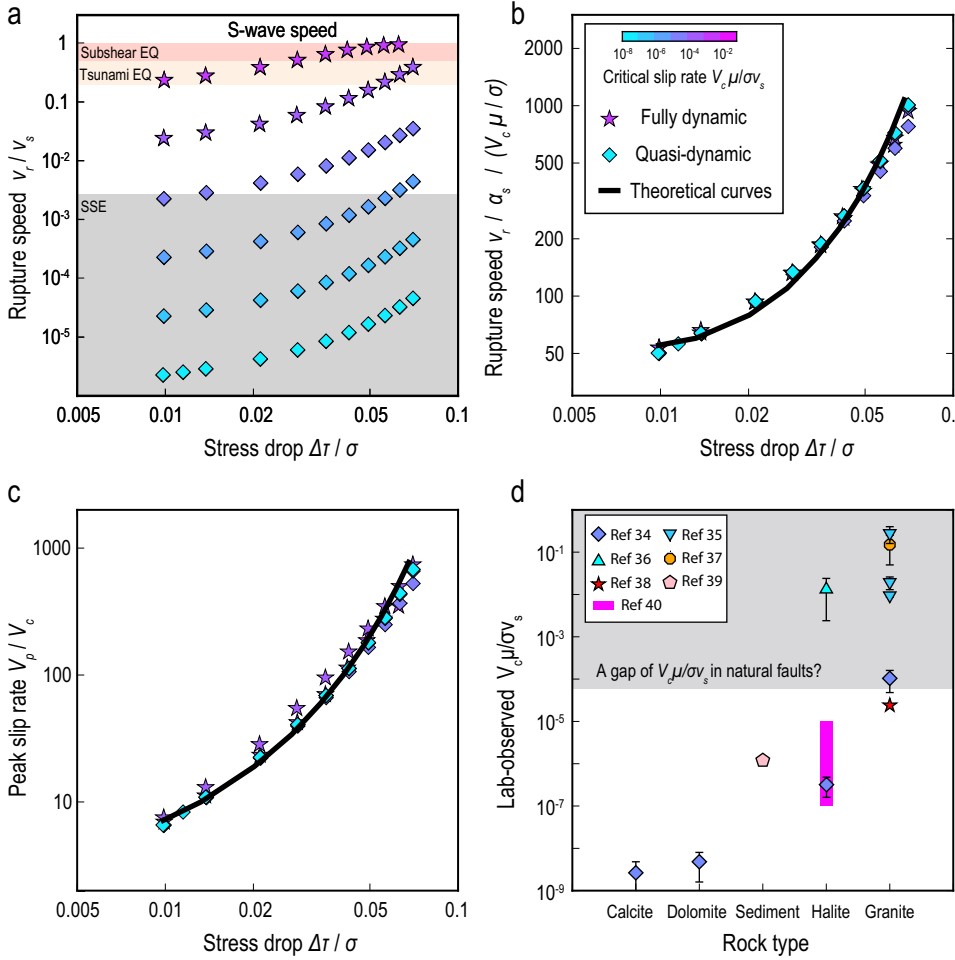

**Fig. 2 | Steady rupture speeds predicted by theory. a** Symbols represent steady rupture speed as a function of stress drop from fully dynamic (stars) and quasi-dynamic (diamonds) simulations, with colour coded by critical slip rate (legend in **b**). $v_s$ and $\sigma$ are the S-wave speed and effective normal stress, respectively. Each symbol represents the result of one single-rupture simulation. **b** Comparison of rupture speeds, divided by the Lorentz contraction factor $\alpha_s$, between numerical simulations (stars and diamonds) and theoretical prediction (black curve). **c** Comparison of peak slip rates between numerical simulations and theoretical prediction. **d** Symbols represent the observed critical slip rates $V_c\mu/\sigma v_s$ in laboratory experiments[35–41] (error bar indicates uncertainty), where $\mu = 40$ GPa and $v_s = 3330$ m/s are assumed. Grey region marks the possible gap of $V_c\mu/\sigma v_s$ in natural faults, whose bottom boundary is approximated based on the rupture speed of the observed "fastest" SSEs, $v_r \sim 10$ m/s.

rupture into a slip pulse (Fig. S6c, d), which is consistent with previous quasi-dynamic simulations[23]. The critical slip rate is an important quantity directly measured in laboratory experiments (Fig. 2d) and such slip-rate-dependent friction behaviour has been suggested to be a universal mechanism for slow slip events[31]. Our numerical simulations show that the steady rupture speed ($v_r/v_s$) can be controlled by two nondimensional parameters (Fig. 2a) related to stress drop ($\Delta\tau/\sigma$) and critical slip rate ($V_c\mu/\sigma v_s$). Here, the quantities are nondimensionalized by the S-wave speed ($v_s$), effective normal stress ($\sigma$) and shear modulus ($\mu$)[58]. The steady rupture speed increases monotonically with the stress drop and critical slip rate. Tuning these two nondimensional parameters produces rupture propagation at a continuum of steady rupture speeds, including speeds of ultra-slow SSEs ($\ll v_s$), tsunami earthquakes[17–19] ($\sim \frac{1}{3}v_s$), and fast subshear earthquakes ($> 0.5v_s$). In all the simulated steady models, $G_c$ agrees with $G_0$ within 3% (Fig. S2a), which validates the steady-state version of equation (1). In addition, $G_c$ increases with $v_r$ (Fig. S2b) while $G_0$ is independent of $v_r$ because, with the friction law adopted here, the static stress drop is controlled by $V_c$ rather than by the peak slip rate (Methods & Fig. S2c).

The numerical simulations further show that the dependence of steady rupture speed on stress drop is highly consistent for various values of critical slip rate, except for the fast subshear ruptures

(Fig. S1b). The fast subshear ruptures deviate from the general trend of the slow ruptures because of dynamic-wave effects. The effects of dynamic waves on rupture propagation have been theoretically investigated[53] and characterised by a nondimensional Lorentz contraction factor, $\alpha_s = \sqrt{1 - (v_r/v_s)^2}$, a well-known function in earthquake dynamics[59]. In addition, fracture mechanics theory shows that the analytical solutions of steady-state ruptures depend on $(v_r/v_s)/\alpha_s$ rather than $v_r/v_s$ (Method). Therefore, the effects of dynamic waves are trivial when $v_r/v_s < 0.5$ (that is $v_r/\alpha_s v_s \approx v_r/v_s$), a speed range including SSEs and tsunami earthquakes, and become significant as $v_r$ approaches $v_s$ (that is $v_r/\alpha_s v_s \to \infty$). Accounting for the Lorentz factor, we find all values of steady rupture speeds, after normalization by $V_c$, collapse onto a universal curve (Fig. 2b), which is predicted by the 3D theory of dynamic fracture mechanics of long ruptures (Methods). All values of peak slip rate also collapse onto the theoretical curve (Fig. 2c). The consistency of these parameters with the 3D theory shows that the propagation of steady ruptures, for the complete spectrum of rupture speeds from arbitrarily slow up to the S-wave speed, can be predicted by the new 3D theory of dynamic fracture mechanics (equation (1)). The validations of equation (1) for steady-state ruptures in this study and for non-steady ruptures in a previous study[53] prove that this equation can be used to describe both SSEs and earthquakes in a same

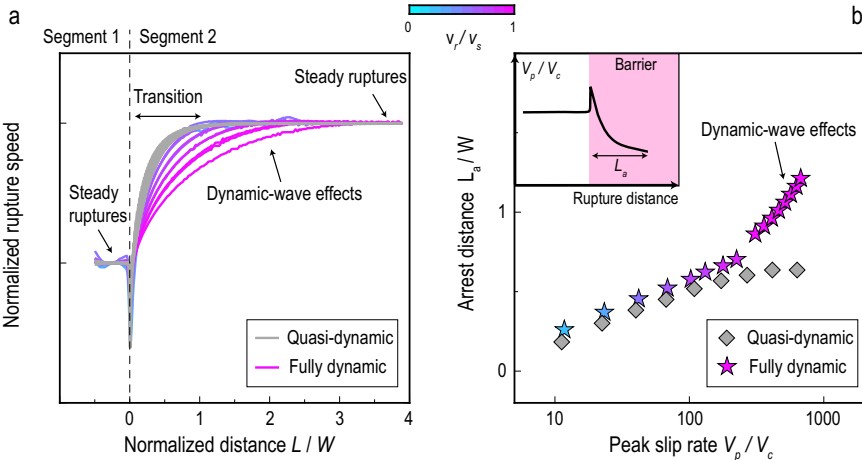

**Fig. 3 | Non-steady ruptures due to along-strike heterogeneities. a** Curves show the transition of rupture speeds from one steady state in segment 1 to another steady state in segment 2 in fully dynamic (coloured curves) and quasi-dynamic (grey curves) simulations. Colour indicates the steady rupture speed in segment 2. **b** Rupture arrest distances inside a barrier versus the peak slip rate before reaching the barrier, from fully dynamic (coloured stars) and quasi-dynamic (grey diamonds) simulations. Colour indicates the steady rupture speed before reaching the barrier. The inset is a sketch of the spatial distribution of rupture speed before and after reaching a barrier.

theoretical framework, provided the dependence of the energy ratio, $G_c/G_0$, on rupture speed, $v_r$, is known. This fundamental theory can be extended in the future to other models with additional complexities, such as fluid effects and off-fault damage.

## Integrated framework for both slow and fast ruptures

$G_c/G_0$ is controlled by the specific frictional behaviour and the resulting stress states of the fault[57]. Because the slip rate and rupture speed are positively correlated (Methods), rate-and-state friction law with rate-strengthening at high slip rates, as observed in laboratory experiments[31–41], produces a V-shaped $G_c/G_0$ as a function of $v_r$ that decreases at low $v_r$ and increases at high $v_r$ (Fig. S2d). Other frictional mechanisms, such as fault gouge dilatancy with an associated change in fluid pressure[27–30], also induce frictional strengthening at high slip rates and may produce a V-shaped $G_c/G_0$. Moreover, fault friction may dramatically decrease at higher seismic slip rates (e.g., > 0.1 m/s) due to flash heating[36–38,60] and thermal pressurization[60,61]. Flash heating, dominant at small earthquake slips, is a rate-weakening mechanism and thus is expected to produce a decreasing $G_c/G_0$ as a function of $v_r$. At larger slips, previous studies suggested[60–63] that the weakening controlled by thermal pressurization or by off-fault inelasticity (bulk plasticity or damage) may be slip-dependent, which remains to be confirmed by future studies accounting for the finite rupture width. Assuming these slip-dependent features, $G_c/G_0$ is a function[42] of the final slip ($D$) independent of $v_r$, $G_c/G_0 \propto D^{n-2}$, where $n = 2/3$ for thermal pressurization[61] and $n = 1$ for off-fault inelastic dissipation[63]. Therefore, considering that $D$ is bounded by the interseismic slip deficit, given a fault coupling map with sufficient resolution[64], the larger the accumulated slip deficit on the fault, the smaller $G_c/G_0$ can be (as a worst-case scenario). Combining the above mechanisms dominant at different slip rates and slips, we propose a conceptual model for $G_c/G_0$ (Fig. 1c) that provides an explanation for the seismological observations[10–13] that show that SSEs may trigger large earthquakes in their adjacent areas. Assuming a rupture starts in a low-coupling fault segment where the rate-strengthening mechanism dominates (due to insufficient slip deficit; purple curve in Fig. 1c), the rupture speed is confined to a low and stable value and thus only forms an SSE. If this steady SSE propagates into the adjacent fault segment where the thermal weakening mechanism dominates (due to sufficient slip deficit; red curves in Fig. 1c), it could transition to a non-steady earthquake and accelerate toward the S-wave speed, which is consistent with 2D cycle simulations with gouge dilatancy and thermal pressurization[65].

Note that there may be two admissible steady speeds if the two competing frictional mechanisms are comparable (dash red curve in Fig. 1c); but we propose that the rupture is more likely to accelerate toward the S-wave speed, which remains to be confirmed in numerical simulations. While the $G_c/G_0$ model presented here is qualitative, it serves as an example of how SSEs and earthquakes can be investigated within the same theoretical framework that combines fracture mechanics theory, laboratory, and seismological observations.

## Along-strike rupture segmentation

Additional results of simulations of non-steady ruptures due to fault heterogeneities demonstrate that non-steady SSEs can also be described by the same theoretical equation of motion. When a steady rupture propagates into a segment of higher shear stress, the rupture jumps from one steady state to another via a transient (Fig. 3a). The rupture speed transients of quasi-dynamic SSEs are very similar to those of dynamic ruptures with $v_r/v_s < 0.5$, while the transition distances of fast ruptures (close to S-wave speed) are quantitatively longer due to the dynamic-wave effects. On the other hand, if the shear stress of the segment is lower than the minimum for steady ruptures, the segment behaves as a barrier so the rupture decelerates and finally arrests after penetrating a certain distance (Fig. 3b). In general, the arresting distance increases with the peak slip rate attained before the rupture reached the barrier, and the arresting distances of fast ruptures are longer due to the dynamic-wave effects.

The reason why fast ruptures have longer transition and arresting distances than slow ruptures can be understood by the 3D theory of dynamic fracture mechanics of long ruptures (equation (1)). The apparent mass, $M(v_r/v_s)$, in the equation of motion is nearly constant when $v_r/v_s < 0.5$ and increases to infinity as $v_r$ approaches $v_s$ (Methods). This is similar to the relativistic mass in Einstein's theory of relativity, which contains the same Lorentz factor with the S-wave speed replaced by the speed of light. Because of this "inertial" effect, a larger apparent mass makes fast ruptures ($v_r \rightarrow v_s$) harder to stop within a barrier or to transition to another steady state, which explains why they require longer transition and arresting distances.

Geophysical observations[8,14,66] show that SSEs usually rupture separate segments of the fault, but some occasionally bridge multiple segments and reach larger magnitudes, which conceptually resembles the supercycle behaviour of large megathrust earthquakes occurring in seismogenic zones[67,68]. This supercycle-like behaviour of SSEs can be explained by the time-dependent evolution of SSE segmentation. Both

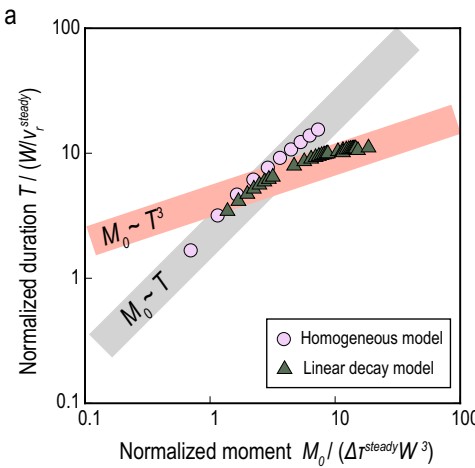

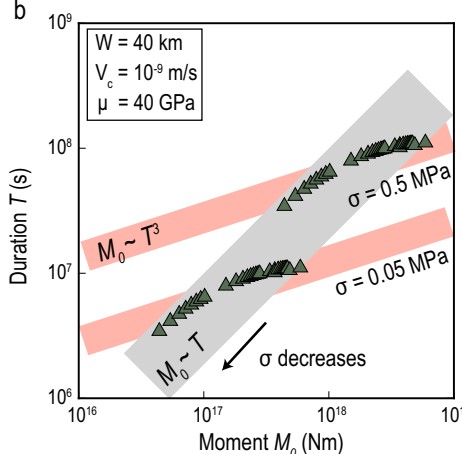

**Fig. 4 | Moment-duration scaling of SSEs controlled by heterogeneities. a** Linear and cubic moment-duration scaling relations based on models with homogeneous shear stress (pink circles) and linearly decaying shear stress (green triangles). $\Delta\tau^{steady}$ and $v_r^{steady}$ are the critical stress drop and rupture speed for steady SSEs, respectively. **b** The arrow shows that the cubic scaling curve is diagonally shifted as effective normal stress $\sigma$ decreases, as predicted by the theory (Methods). A linear envelope of moment-duration scaling (grey region) is shown assuming diverse values of $\sigma$.

the numerical simulations and theory demonstrate that there is a critical stress drop for steady SSEs (Fig. 2b & Methods), which is $\Delta\tau^{steady}/\sigma \approx 0.01$ for the rate-and-state frictional parameters used in this paper. On dip-slip faults, a critical final slip is approximately related to the critical stress drop[42] by $D^{steady} = 2W\Delta\tau^{steady}/\pi\mu$, where $W$ is the SSE rupture width. Fault segments need to accumulate sufficient slip deficit, that is $D^{steady} = 0.02W\sigma/\pi\mu$, to be capable of accommodating steady SSE ruptures, otherwise they act as barriers to stop rupture propagation. Therefore the recurrence interval of steady ruptures can be estimated by the ratio of $D^{steady}$ to the slip deficit rate on the fault segments during the inter-SSE period. The observed slip deficit rates on SSE fault segments in subduction zones globally are diverse, ranging from < 10% up to > 50% of the plate convergence rate[8,69–71], which can be explained by different values of fault properties in cycle simulations[72]. As rough lower bound estimates, values of $\sigma \sim 0.1 - 10$ MPa, $W \sim 40$ km, $\mu \sim 40$ GPa, and 100% coupling at a plate convergence rate of $10^{-9}$m/s yield $\Delta\tau^{steady} \sim 0.001 - 0.1$ MPa and recurrence times of $\sim 0.2 - 20$ months, which are comparable to the typical stress drops of $0.001 - 0.2$ MPa[14,46,73] and typical recurrence times of months to years[8,74] of SSEs globally.

The recurrence intervals of large megathrust earthquakes[68] are much longer than those of SSEs. Laboratory experiments[62] and theoretical models[60,61] of thermal weakening show that rock friction exponentially decays from a peak to a residual value, resulting in an upper bound estimate[62] of fracture energy: $G_c \approx A(\mu_p - \mu_r)\sigma^{1-B}$, where $\mu_p$ and $\mu_r$ are the peak and residual friction coefficients, respectively, and the coefficients $A$ and $B$ can be experimentally determined. The energy release rate on a long fault assuming a constant stress drop is $G_0 = \Delta\tau^2 W/\pi\mu$ (Methods), therefore the critical stress drop for runaway earthquake ruptures, derived from the condition $G_c/G_0 < 1$, is $\Delta\tau^2/\sigma^{1-B} \approx A\pi(\mu_p - \mu_r)\mu/W$. Given $A \approx 3 - 78$, $B \approx 1.18$, and $\mu_p - \mu_r \approx 0.7$ in laboratory experiments[62], and $W \sim 40$ km and $\mu \sim 40$ GPa, we have $\Delta\tau^{run} \approx (2.6 - 13)\sigma^{-0.09}$, which is virtually insensitive to $\sigma$ because the associated exponent is close to zero. Note that $\Delta\tau^{run}$ is an upper bound estimate. For $\sigma > 20$ MPa, $\Delta\tau^{run} \approx 2 - 10$ MPa is comparable to the average stress drop of $\sim 4$ MPa of global large earthquakes[75]. The stress drop of large earthquakes is close to $\Delta\tau^{run}$ if runaway occurs at the first possible moment. Then the fault strength during the interseismic period increases from the residual value $\sigma\mu_r$ to $\sigma\mu_r + \Delta\tau^{run}$, resulting in an average apparent fault strength of $\sigma\mu_r + 0.5\Delta\tau^{run}$ given a constant stressing rate. Note that the apparent fault strength is derived based on the assumption of the thermal weakening with exponential

slip-weakening as observed in laboratory experiments[62] and the complexity of fault roughness[76] is not considered. The apparent fault strength is much smaller than the static fault strength of earthquake ruptures ($\sigma\mu_p$) when $\sigma > 50$ MPa, that is $\mu_r + 0.5\Delta\tau^{run}/\sigma \ll \mu_p$, which can explain why some crustal faults are much weaker than the predicted static fault strength[77]. As a rough lower bound estimate of the recurrence interval, 100% coupling at a plate convergence rate of $10^{-9}$ m/s yields $\sim 40 - 200$ years, which is comparable to the recurrence times of tens to hundreds years of large earthquakes[68]. Large earthquake supercycles are challenging to study due to their long recurrence times, but the formal connections between regular earthquakes and SSEs revealed here indicate that investigations of the kinematics and dynamics of frequent SSEs leading to a comprehensive SSE supercycle model can advance our understanding of the supercycle behaviour of large devastating earthquakes.

## Observations of SSEs and earthquakes

The comparison of moment-duration scaling relations between SSEs and earthquakes has been considered in discussions of their physical mechanisms[14,45–52]. However, the moment-duration relation of SSEs observed in a particular environment features a cubic scaling[14] that is radically different from the linear scaling observed in global compilations from different fault environments[45]. Here, we show that the debated scaling behaviours can be attributed to different types of fault heterogeneities: heterogeneity of shear stress can produce a cubic scaling, whereas heterogeneity of effective normal stress produces a linear scaling. The relation between moment ($M_0$) and duration ($T$) is $M_0 \propto \Delta\tau W^2 L$ for long ruptures[78], where $L = v_r T$ is the rupture length and $\Delta\tau$ and $v_r$ are the stress drop and rupture speed, respectively. Defining $\Delta\tau \propto L^\alpha$ and $v_r \propto L^\beta$ leads to $M_0 \propto T^{\frac{1+\alpha}{1-\beta}}$, where $\alpha$ and $\beta$ are constant coefficients. For a homogeneous model (Methods), ruptures with different values of uniform $\Delta\tau$ produce a linear moment-duration scaling (Fig. 4a). However, if the shear stress distribution on the fault is heterogeneous, and in particular if it decays linearly away from the nucleation area (Methods), the simulated models result in $\alpha = 0.5$ and $\beta = 0.5$, which leads to a cubic scaling relation (Fig. 4a & S3) consistent with the observations in the Cascadia subduction zone (Fig. S3c & S3d). Although this is one specific case of heterogeneity, it demonstrates that a cubic scaling relation can be produced by heterogeneity of shear stress within a particular fault, as also observed in an SSE cycle model[52]. Moreover, such a cubic scaling curve, obtained under the assumption of constant effective normal stress $\sigma$, is diagonally shifted

if $\sigma$ varies over a significant range, in a way that produces a linear $M_0 \propto T$ envelope (Fig. 4b). This is predicted by the theoretical relations $M_0 \propto \Delta\tau \propto \sigma$ and $T \propto 1/v_r \propto \sigma$ (Methods), which imply $M_0 \propto T$ when only $\sigma$ is varied. Therefore, when mixing data with diverse values of $\sigma$, a linear envelope of the $M_0 - T$ scaling, different from that of regular earthquakes, can be obtained. This can explain the observed linear scaling based on a global compilation of various slow earthquakes from different fault environments whose $\sigma$ may span several orders of magnitude, such as $\sigma \sim 1 - 10$ kPa on the deep San Andreas fault[79] and $\sigma > 1$ MPa in the Izu-Bonin Trench[16]. The explanation based on the diverse values of $\sigma$ is also supported by the diverse values of stress drop of the observed SSEs, which span at least 3 orders of magnitude (Fig. S3c). None of the other parameters of the model produce a linear envelope when varied, which would require $M_0$ and $T$ to have the same scaling dependence on that parameter.

To explore the scaling relation of rupture speeds that can be directly compared with fracture mechanics theory, we compiled and calculated the rupture speeds and peak slip rates of global SSEs, earthquakes and laboratory ruptures (Fig. 1b & Methods). In general, the observed rupture speed increases with the observed peak slip rate, enveloped by two theoretical predictions with peak-to-residual strength drops of $\Delta\tau_{p-r}/\mu = 10^{-6}$ and $\Delta\tau_{p-r}/\mu = 10^{-4}$ at the rupture tip (Fig. 1b). Assuming the stress drops of SSEs are approximately the minimum value of steady ruptures, as suggested by cycle simulations[23], the effective normal stress of the SSEs can be constrained by the theoretical relation, $\sigma \approx \Delta\tau_{p-r}/0.07$ (Methods). Given $\mu = 40$ GPa, the constrained $\sigma$ of the SSEs and lab experiments lies between $\sim 0.6$ and $60$ MPa. In addition, another theoretical relation, $V_c/\sigma \approx v_r^{steady}/50\mu$ (Methods), can constrain the critical slip rate: $V_c \sim 3 \times 10^{-9} - 3 \times 10^{-4}$ m/s, which is consistent with the reported values $10^{-9} - 10^{-2}$ m/s in laboratory experiments[35–41]. Note that the uncertainty in constraining $V_c$ alone is larger than that for the ratio $V_c/\sigma$, because the former requires estimates of both rupture speed and peak slip rate whereas the latter only requires estimates of rupture speed.

Although a continuum of rupture speeds has been reported in laboratory experiments[22,34], there exists an observation gap of rupture speeds in natural faults between $10 - 1000$ m/s (Fig. 1b). The numerical models show that a continuum of rupture speeds can be obtained by tuning two nondimensional parameters, $\Delta\tau/\sigma$ and $V_c\mu/\sigma v_s$ (Fig. 2a). Conversely, a gap of rupture speeds is expected if $\Delta\tau/\sigma$ and $V_c\mu/\sigma v_s$ are not sufficiently diverse. SSE cycle simulations[23] showed that $\Delta\tau/\sigma$ of SSEs is close to the minimum value of steady ruptures $\Delta\tau^{steady}/\sigma$, and thus it is reasonable to assume that rupture speed is mainly controlled by $V_c\mu/\sigma v_s$, that is $v_r/v_s \approx 50V_c\mu/\sigma v_s$ (Methods). Here, we propose two explanations for the gap of rupture speeds in natural faults. The first one is that the maximum $V_c\mu/\sigma v_s$ in natural faults is much smaller than the maximum value reported in laboratory experiments on natural and synthetic fault gouges (Fig. 2d). If $V_c\mu/\sigma v_s < 10^{-4}$ in natural faults then $G_c/G_0$ is expected to be an increasing function of $v_r$ for $v_r < 10$ m/s and a decreasing function for $v_r > 10$ m/s. The former produces steady SSEs with $v_r < 10$ m/s (grey area in Fig. 1c) and the latter produces non-steady earthquakes with an average rupture speed of $v_r > 1000$ m/s (pink area in Fig. 1c). The second explanation is that values of $V_c\mu/\sigma v_s > 10^{-4}$ are possible in natural faults (grey area overlaps with pink area in Fig. 1c), but are very rare so that we have not yet observed events within the intermediate speed range. This explanation can be justified by observations in laboratory experiments[22] that show that events with rupture speeds between 10 and 1000 m/s are rarer than slower ruptures. However, more work is needed either to explain why $V_c\mu/\sigma v_s$ in natural faults is not as diverse as observed in laboratory experiments or to fill the observational gap of rupture speeds in natural faults, by capitalizing on advances in rock mechanics, geodesy and seismology.

Earthquake ruptures on long faults can steadily propagate at speeds higher than $\frac{1}{3}v_s$ if $V_c\mu/\sigma v_s > 6 \times 10^{-3}$, which provides a possible new mechanism to explain the anomalously slow tsunami earthquakes[17–19,80]. Given $\sigma = 20$ MPa, $\mu = 40$ GPa and $v_s = 3330$ m/s, values of $V_c$ are required to be larger than $10^{-2}$ m/s, which is the upper bound of the observed values in laboratory experiments[35–41]. As the frictional strength may change from rate-strengthening to rate-weakening at slip rates higher than 0.1 m/s due to thermal weakening mechanisms that facilitate fast earthquakes[36–38], the narrow range of rate-strengthening behaviour between $10^{-2}$ and 0.1 m/s may explain the scarcity of tsunami earthquakes. Alternative explanations for tsunami earthquakes are low rigidity materials[81] and inelastic deformation around the fault[82], and the density and size of asperities[83], which remain to be confirmed by further investigations.

## Discussion

While rock frictional behaviour may be controlled by different mechanisms, the rupture propagation of both slow and fast events on long faults can be predicted by the same 3D theory of dynamic fracture mechanics and the rupture speed is quantitatively controlled by $G_c/G_0$. Integrating laboratory and theoretical developments of frictional mechanisms in quantifying $G_c/G_0$, this basic model would enable a quantitative description of both the short-term slow ruptures and the long-term supercycle behaviours associated with large earthquakes within the same theoretical framework. This fundamental model integrates slow and fast ruptures, reconciles the debated scaling relations, and thus opens new avenues for assessing the future seismic hazard through integration of observations and models of frequently occurring SSEs and devastating earthquakes.

## Methods

### Estimates of rupture speed and peak slip rate

To explore a universal scaling relation in global SSEs, earthquakes and laboratory experiments, we compiled datasets of events from the literature[14,16–19,22] and online databases[15,20,21] and calculated their rupture speed and peak slip rate. For SSEs and earthquakes, we estimated rupture speeds as $v_r = L/T$, with an uncertainty of a factor of 2 for bilateral ruptures, where $L$ and $T$ are the rupture length and duration, respectively. The peak slip rate is estimated by $V_p = \gamma\bar{D}/\tau_{rise}$, where $\bar{D}$ is the average slip, $\tau_{rise}$ is the rise time, and $\gamma$ is an empirical ratio that links the peak and average slip rates. For ruptures with aspect ratios $\geq 1$, the rise time is approximately estimated by $\tau_{rise} = TW/L$, where $W$ is the rupture width. $L$, $W$, $\bar{D}$, and $T$ of SSEs are constrained by geodetic observations[14–16]. $L$, $W$, and $\bar{D}$ of earthquakes are constrained by finite fault rupture models[17–20]; $T$ of three tsunami earthquakes is constrained by finite fault rupture models[17–19] and $T$ of other earthquakes is constrained by source time functions[21]. $\gamma \approx 20$ is assumed for both SSEs and earthquakes based on the results of numerical simulations (Fig. S4). The uncertainty of $\gamma$ affects the values of peak slip rate, but not the values of rupture speeds. The rupture speed in laboratory experiments[22] is directly measured from strain gauge array data and the peak slip rate is estimated based on the direct measurements of dynamic stress drop and rupture speed.

### General conditions for steady ruptures on long faults

For subshear mode III (dip-slip), rupture propagation on long faults with finite width ($W$) in a 3D elastic medium can be predicted by a theoretical rupture-tip equation of motion[53]

$$F(G_c/G_0) = M(v_r) \cdot \dot{v}_r, \tag{3}$$

where

$$F(G_c/G_0) = 1 - G_c/G_0,$$
$$M(v_r) = \frac{W}{v_s^2}\frac{\gamma}{A\alpha_s^P}, \tag{4}$$

$G_c$ and $G_0$ are the fracture energy and steady-state energy release rate, $v_r$ is the rupture speed, $\dot{v}_r$ is the rupture acceleration, $v_s$ is the S-wave speed, $\alpha_s = \sqrt{1-(v_r/v_s)^2}$ is the Lorentz contraction factor, and $\gamma$, $A$, and $P$ are known coefficients[53]. In general, this equation of motion can be expressed as

$$F\left(\frac{G_c(v_r)}{G_0(v_r)}\right) = M(v_r) \cdot \dot{v}_r, \tag{5}$$

where $G_0(v_r)$ and $G_c(v_r)$ are functions of $v_r$ that depend on the specific friction law considered[57].

For steady-state ruptures, $\dot{v}_r = 0$, thus the first condition for steady ruptures is

$$G_c(v_r) = G_0(v_r). \tag{6}$$

For the stability analysis, we define

$$f(\dot{v}_r, v_r) \equiv \frac{1 - M(v_r)\dot{v}_r}{G_c(v_r)/G_0(v_r)}, \tag{7}$$

where the function $f$ can be written as a linear expansion near the steady-state solution $\dot{v}_r = 0$

$$f(\dot{v}_r, v_r) \approx f(0, v_r) + \frac{\partial f(0, v_r)}{\partial \dot{v}_r} d\dot{v}_r + \frac{\partial f(0, v_r)}{\partial v_r} dv_r = 1. \tag{8}$$

If a positive perturbation of $v_r$ is applied to the steady-state rupture tip, that is $dv_r > 0$, the rupture can be stable only when the response of $\dot{v}_r$ is negative, that is $d\dot{v}_r < 0$. Therefore the stability condition for a steady rupture is

$$\frac{\partial f}{\partial \dot{v}_r} \cdot \frac{\partial f}{\partial v_r} > 0. \tag{9}$$

Because $\partial f / \partial \dot{v}_r \propto -M(v_r) < 0$, a steady-state rupture requires

$$\frac{\partial f(0, v_r)}{\partial v_r} < 0. \tag{10}$$

Since $G_c(v_r) > 0$ and $G_0(v_r) > 0$, we have

$$\frac{d(G_c/G_0)}{dv_r} > 0. \tag{11}$$

Considering $f(0, v_r) = 1$, equation (11) yields

$$\frac{dG_c(v_r)}{dv_r} > \frac{dG_0(v_r)}{dv_r}. \tag{12}$$

Some 2D analog laboratory experiments[22,34,43,44] have suggested a continuum of rupture speeds. However, to our knowledge, there are no laboratory experiments yet on long ruptures with zero-slip conditions beyond a finite width $W$ that could have addressed our problem. The conditions for steady-state rupture in existing experiments, without $W$ effect, are radically different. In existing 2D analog experiments, the energy release rate is a function of rupture speed, rupture length and stress drop: $G(v_r, L, \Delta\tau)$. Because this $G$ does not depend on $dv_r/dt$ (a property often described as "the crack tip has no inertia"), without $W$ effect the steady-state condition cannot be obtained by setting $dv_r/dt = 0$ into the equation of motion. Moreover, $G_c$ needs to be proportional to rupture size $L$ to compensate the linear growth of $G$ with $L$ in order to maintain a steady rupture speed $v_r$. In addition, because the dependency of $G$ on $v_r$ is negligible when $v_r \ll v_s$, very fine tuning of $dG_c/dL$ is required to obtain a variety of steady slow rupture

speeds. Therefore, in the absence of the $W$ effect, steady ruptures can occur only under restricted mathematical conditions. Those features of previous theories and experiments are in striking contrast to the results of our modelling accounting for the $W$ effect.

## Quasi-dynamic SSE rupture simulations

We consider a 3D dip-slip rupture problem on an infinitely long fault with finite seismogenic width, $W$, embedded in a full-space, linear elastic, homogeneous medium. This 3D elongated rupture problem has been successfully approximated by a reduced dimensionality (2.5D) model, which accounts for the elongated features while having a low computational cost[23,53]. To facilitate a comprehensive comparison between numerical simulations and fracture mechanics theory, we investigate the rupture propagation of SSEs and earthquakes using 2.5D single-rupture simulations with prescribed initial conditions. The simulations of SSEs are quasi-dynamic, while the simulations of earthquakes are fully dynamic. The shear modulus and S-wave speed of the medium are denoted $\mu$ and $v_s$, respectively.

The frictional strength, $\tau$, of faults is assumed to be controlled by a rate-and-state friction law with rate-weakening behaviour at low slip rates and rate-strengthening behaviour at high slip rates[84], which has been used to investigate the rupture propagation of SSEs[23–25,85]

$$\tau = f^*\sigma + a\sigma \ln\left(\frac{V}{V^*}\right) + b\sigma \ln\left(\frac{V_c\theta}{D_c} + 1\right), \tag{13}$$

where $\sigma$ is the effective normal stress, $f^*$ and $V^*$ are arbitrary reference values, $D_c$ is the characteristic slip distance, $a$ and $b$ are nondimensional parameters, $V$ is the slip rate, $\theta$ is the state variable, and $V_c$ is a critical slip rate. A fault exhibits rate-weakening frictional behaviour when $a - b < 0$, and the critical slip rate $V_c$ controls the transition from rate-weakening to rate-strengthening[23] (Fig. S6a). The evolution of state $\theta$ is described by the aging law[86]

$$\dot{\theta} = 1 - \frac{V\theta}{D_c}, \tag{14}$$

where $\dot{\theta}$ is the time derivative of $\theta$.

For each single-rupture model, one of the primary parameters that affects the rupture propagation is the initial shear stress $\tau_i$, which is prescribed by the values of initial slip rate $V_i$ and state $\theta_i$

$$\tau_i = f^*\sigma + a\sigma \ln\left(\frac{V_i}{V^*}\right) + b\sigma \ln\left(\frac{V_c\theta_i}{D_c} + 1\right), \tag{15}$$

The nondimensional parameters, $a/b$ and $W/L_c$, also affect the rupture propagation[23], where the size of the weakening process zone is

$$L_c = \frac{\mu D_c}{b\sigma}. \tag{16}$$

In this study, we fix the nondimensional ratios of $a/b = 0.8$ and $W/L_c = 400$, and systematically vary $\tau_i$ and $V_c$. The specific values of the frictional parameters are prescribed as: $\sigma = 20$ MPa, $b = 0.015$, $W = 40$ km, $D_c = 10^{-3}$ m, $f^* = 0.6$ and $V^* = 10^{-9}$ m/s. The choice of these values does not affect the conclusions of this paper, because both the computational and analytical results are presented in nondimensional form. The increase in fault stress due to plate convergence is small during the short time of rupture propagation and thus we ignore it.

We smoothly nucleate unilateral ruptures by prescribing a nucleation zone of length $0.5W$ with higher slip rates ($\geq 10V_c$), which slowly loads its surrounding region. Outside the nucleation zone rupture propagation is spontaneous. A stronger nucleation, such as the overstressed nucleation condition, results in slight oscillations of rupture speed in the fully dynamic rupture models, but does not affect

the steady rupture speed (Fig. S5). We use the boundary element software QDYN[87] for the quasi-dynamic SSE simulation, where the fault is infinitely long and the fault slip is horizontally periodic with a prescribed length of $11W$. To avoid the interaction of the periodic fault segments, we set a buffer segment of length $5.5W$ with rate-strengthening frictional behaviour ($a > b$). Sufficient numerical resolution is guaranteed by setting a small grid size $\Delta x = L_c/8$. We set the simulated time long enough to capture the whole rupture propagation. For each single-rupture model, we determine the rupture time on each fault node by a slip rate threshold of $10V_i$, and compute the rupture speed as the inverse of the along-strike gradient of the rupture time.

### Fully dynamic earthquake rupture simulations

We conducted 2.5D fully-dynamic single-rupture simulations of earthquakes with the spectral element software SEM2DPACK[88]. For a quantitative comparison between SSE and earthquake simulations, the same friction law and parameters are assumed in the dynamic earthquake rupture simulations, except for larger values of $V_c$ and the additional thermal weakening[60,89] at slip rate > 0.1 m/s is not considered. Previous theoretical studies[53] have suggested that thermal weakening can affect the rupture speeds by controlling the dissipated[61] and potential[89] energies on faults, which remains to be quantitatively investigated in 2.5D in the future. For simulations with rupture speeds close to the S-wave speed, we set a sufficiently large computational domain to avoid effects of waves reflected from domain boundaries within the simulated time. For simulations with slow rupture speeds, the seismic radiation is relatively weak and can be well absorbed by the default absorbing boundaries in SEM2DPACK, and therefore, the simulated times are allowed to be several times longer than those for fast rupture speeds. We set the time step based on the Courant-Friedrichs-Lewy stability condition and the grid size is the same as in the quasi-dynamic SSE simulations, $\Delta x = L_c/8$.

### Energy balance of steady SSEs and earthquakes

For SSE and earthquake ruptures on long faults with finite width $W$, the energy release rate and dissipated fracture energy can be derived in the theoretical framework of 3D dynamic fracture mechanics. The steady energy release rate $G_0$ is the rate of mechanical energy flow into the rupture tip per unit rupture advance for steady ruptures. For dip-slip faulting, $G_0$ depends on the static stress drop ($\Delta\tau$) and fault width:

$$G_0 = \frac{\lambda \Delta\tau^2 W}{\mu}, \tag{17}$$

where $\lambda$ is a geometrical factor, with $\lambda = 1/\pi$ for a deeply buried fault[53]. The energy release rate is dissipated by fracture energy $G_c$, which depends on the strength evolution on the fault[90]:

$$G_c = \int_0^D [\tau(\delta) - \tau(D)] \cdot d\delta, \tag{18}$$

where $\tau(\delta)$ is the fault strength as a function of fault slip, $\delta$, and $D$ is the final slip. Equations (17) and (18) are the generic definitions of energies of ruptures on long faults, regardless of the specific friction law. Below, we propose an approach to estimate $G_0$ and $G_c$ under the framework of the V-shape rate-and-state friction law.

$G_0$ is a function of the static stress drop, the difference of shear stress before and after the ruptures

$$\Delta\tau = \tau_i - \tau_f, \tag{19}$$

where $\tau_i$ are $\tau_f$ are the initial shear stress and final shear stress, respectively. Rupture simulations of V-shape rate-and-state friction[23]

show that the fault strength approximately drops to the minimum steady-state strength within a distance from the rupture tip shorter than $W$ (Fig. S6c, d) and stays there until the end of the rupture, which is a feature different from the regular rate-and-state friction with aging law[91]. The minimum steady-state strength[23] is

$$\tau_f = f^* \sigma + a\sigma \ln\left(\frac{b-a}{a}\frac{V_c}{V^*}\right) + b\sigma \ln\left(\frac{a}{b-a}+1\right). \tag{20}$$

Combining equations (15) and (19) with (20) yields the following close-form expression of static stress drop:

$$\Delta\tau = a\sigma \ln\frac{aV_i}{(b-a)V_c} + b\sigma \ln\frac{\frac{V_c\theta_i}{D_c}+1}{\frac{a}{b-a}+1}. \tag{21}$$

Equation (21) predicts the numerical values of $\Delta\tau$ well in all the simulated steady models (Fig. S2c). Substituting equation (21) into equation (17) yields the theoretical energy release rate

$$G_0 = \frac{\lambda b^2 \sigma^2 W}{\mu} \cdot \left[\frac{a}{b}\ln\frac{aV_i}{(b-a)V_c} + \ln\frac{\frac{V_c\theta_i}{D_c}+1}{\frac{a}{b-a}+1}\right]^2. \tag{22}$$

The main feature in equation (22) is that $G_0$ only depends on the prescribed parameters and is independent of the peak slip rate $V_p$. As only $\tau_i$ and $V_c$ are systematically varied in this study, $G_0$ can be written as $G_0(\tau_i, V_c)$.

$G_c$ is an integral function of the fault strength $\tau(\delta)$ over the slip $\delta$. The numerical simulations show that fault strength governed by V-shape rate-and-state friction has two weakening stages: the first stage accounts for the fast weakening process and the second stage accounts for the slow weakening process (Fig. S6b). This friction law produces a persistent and non-linear slip weakening similar to that caused by the lab-observed thermal weakening[92], which obviously differs from the nearly-linear slip weakening produced by the regular rate-and-state friction with the aging law[91]. The resulting ruptures are pulse-like and we find that the slip rate in the tail of the pulse decays exponentially with distance (Fig. S6e, f), which is fundamentally different than the power law decay that signals unconventional singularities in the 2D models by Brener and Bouchbinder[93], even with a similar friction law. In the first weakening stage, the strength drop, $\Delta\tau_{p-r}$, and the critical slip-weakening distance, $d_c$, can be predicted well by the theoretical equations[23]

$$\begin{aligned}\Delta\tau_{p-r} &= b\sigma \left[\ln\left(\frac{V_c\theta_i}{3D_c}+1\right) - \ln\left(\frac{3V_c}{V_p}+1\right)\right], \\ d_c &= D_c \left[\ln\left(\frac{V_c\theta_i}{3D_c}+1\right) - \ln\left(\frac{3V_c}{V_p}+1\right)\right], \end{aligned} \tag{23}$$

where $V_p$ is the peak slip rate and the factor 3 is an approximation of the non-uniform slip rate within the first weakening stage, which was proposed to be 2 by Hawthorne and Rubin[23]. Thus, we estimate the fracture energy caused by the first weakening stage as

$$G_{c1} = \frac{1}{2} d_c \Delta\tau_{p-r} = \frac{1}{2} b\sigma D_c \left[\ln\left(\frac{V_c\theta_i}{3D_c}+1\right) - \ln\left(\frac{3V_c}{V_p}+1\right)\right]^2. \tag{24}$$

The contribution of fracture energy of the second weakening stage also needs to be considered. Here, we account for this part of the total fracture energy by

$$G_{c2} = \frac{1}{2}(d_c + D)(\tau_r - \tau_f), \tag{25}$$

where $D$ is the final slip, $\tau_f$ is the final fault strength after rupture arrest and $\tau_r$ is the fault strength at the tail of the first weakening stage

$$\tau_r = f^* \sigma + a\sigma \ln\left(\frac{V_p}{3V^*}\right) + b\sigma \ln\left(\frac{3V_c}{V_p} + 1\right), \tag{26}$$

$$\tau_r - \tau_f = a\sigma \ln\left(\frac{aV_p}{3(b-a)V_c}\right) + b\sigma \ln\left(\frac{\frac{3V_c}{V_p}+1}{\frac{a}{b-a}+1}\right). \tag{27}$$

For ruptures on long faults with finite width $W$, the final slip, $D$, is proportional to the static stress drop, $\Delta\tau$, that is[42]

$$D = \frac{2\lambda W}{\mu} \cdot \Delta\tau. \tag{28}$$

Substituting equations (21), (28), (23), and (27) into equation (25) yields the close-form expression of the second part of the fracture energy, $G_{c2}$. The close-form expression of the total fracture energy is given by $G_c = G_{c1} + G_{c2}$. As $G_c$ depends on $\tau_i$, $V_c$, and the undetermined peak slip rate $V_p$, it can be written as $G_c(V_p, \tau_i, V_c)$.

For steady ruptures, the energy release rate shall be balanced by the dissipated fracture energy:

$$G_c(V_p, \tau_i, V_c) = G_0(\tau_i, V_c). \tag{29}$$

Equation (29) shows that the peak slip rate, $V_p$, of steady ruptures can be uniquely determined from the energy balance condition of V-shape rate-and-state friction. We find that equation (29) predicts the relations among $V_p$, $\Delta\tau$, $G_0$, and $G_c$ well in all the simulated steady ruptures (Fig. 2c & S2), which demonstrates that the two weakening stages shall be considered as a whole cohesive zone in calculating the fracture energy. The steady rupture speed, $v_r$, monotonically depends on $V_p$ (equation (34)), thus equation (29) can also be written as:

$$G_c(v_r; \tau_i, V_c) = G_0(\tau_i, V_c). \tag{30}$$

### Relation between peak slip rate and rupture speed

A linear relation between peak slip rate and rupture speed for steady SSEs has been proposed by Hawthorne and Rubin[23]

$$V_p = \frac{v_r}{C} \cdot \frac{\Delta\tau_{p-r}}{\mu}, \tag{31}$$

where $C \approx 0.5 - 0.55$ is an empirical geometrical factor. But this relation does not include the effects of dynamic waves when the rupture speed approaches the S-wave speed. Alternatively, Gabriel et al.[63] have provided a theoretical relation between peak slip rate and rupture speed for 2D strike-slip faulting earthquakes whose rupture speeds are close to the S-wave speed. Here, we extend their 2D strike-slip relation to a dip-slip relation for the 3D long rupture problem, which physically incorporates equation (31), as explained below.

Weng and Ampuero[53] demonstrated that if the cohesive zone size on long faults is much smaller than fault width, $L_c \ll W$, then the energy release rate has the following form:

$$G_{tip} = \frac{1}{2\mu} A(v_r) K_{tip}^2, \tag{32}$$

where $A(v_r) = 1/\alpha_s$, $\alpha_s = \sqrt{1 - (v_r/v_s)^2}$ is the Lorentz contraction term and $K_{tip}$ is the stress intensity factor. By removing the strike-slip term $1 - \nu$ and replacing $A(v_r)$ by $1/\alpha_s$ in equation (18) in Gabriel et al.[63], we obtain the dip-slip relation between peak slip rate and rupture speed,

similar to a classical 2D result[94]

$$V_p = \frac{v_r}{\alpha_s} \cdot \frac{2\Delta\tau_{p-r}}{\mu}, \tag{33}$$

where the correction of a factor of 2 is made to fit the numerical results. If $v_r \ll v_s$, then the Lorentz term $\alpha_s = 1$, and equation (33) is the same as equation (31) proposed for SSEs by Hawthorne and Rubin[23]. Note that $\Delta\tau_{p-r}$ is a function of $V_p/V_c$ (equation (23)), and thus equation (33) can be written as

$$\frac{V_p}{V_c} = \frac{v_r/\alpha_s v_s}{V_c \mu/v_s \sigma} \cdot \frac{2\Delta\tau_{p-r}}{\sigma}, \tag{34}$$

Equation (34) describes the relation between peak slip rate and rupture speed. By combining equations (34) and (29), we obtain a relation between stress drop and rupture speed for both SSEs and earthquakes (Fig. 2).

### Critical stress drop and strength drop for steady ruptures

Here we derive the critical stress drop for steady ruptures, $\Delta\tau^{steady}/\sigma$, under a V-shape rate-and-state friction law, where $G_0$ and $G_c$ can be written as (above equations)

$$\begin{aligned} G_0 &= \frac{\lambda W}{\mu}\Delta\tau^2, \\ G_c &= \frac{\lambda W}{\mu}\left[\frac{L_c}{2\lambda W}\Delta\tau_{p-r}^2 + \left(\frac{L_c}{2\lambda W}\Delta\tau_{p-r} + \Delta\tau\right)(\tau_r - \tau_f)\right]. \end{aligned} \tag{35}$$

For steady ruptures, the energy balance condition $G_0 = G_c$ based on equation (35) is

$$\left(\frac{\Delta\tau}{b\sigma}\right)^2 = \frac{L_c}{2\lambda W}\left(\frac{\Delta\tau_{p-r}}{b\sigma}\right)^2 + \left(\frac{L_c}{2\lambda W}\frac{\Delta\tau_{p-r}}{b\sigma} + \frac{\Delta\tau}{b\sigma}\right)\frac{\tau_r - \tau_f}{b\sigma}. \tag{36}$$

It is possible to analytically solve for $\Delta\tau^{steady}/b\sigma$ in equation (36), however the resulting expression is complex and lengthy and does not provide additional physical insight, therefore we only solve it numerically. Considering $W/L_c \gg 1$, equation (36) can be simplified approximately as

$$\frac{\Delta\tau^{steady}}{b\sigma} = \frac{\tau_r - \tau_f}{b\sigma} = \frac{a}{b}\ln\left(\frac{aV_p}{3(b-a)V_c}\right) + \ln\left(\frac{\frac{3V_c}{V_p}+1}{\frac{a}{b-a}+1}\right). \tag{37}$$

Hawthorne and Rubin[23] noted that the critical stress drop for steady ruptures can be estimated assuming $V_p/V_c \approx 15(b-a)/a$. Here, we use the approximate value of $V_p/V_c \approx 30(b-a)/a$ and calculate $\Delta\tau^{steady}/b\sigma$ from equations (36) and (37). Both solutions can explain the simulation results with an uncertainty of a factor of 2 (Fig. S7). Given the values of $a/b = 0.8$, $b = 0.015$, and $W/L_c = 400$ used in this paper, the critical stress drop for steady ruptures is about $\Delta\tau^{steady}/\sigma \approx 0.01$. Substituting these values into equations (23) and (34) yields $v_r^{steady} \approx 50\alpha_s V_c \mu/\sigma$ and $\Delta\tau_{p-r}^{steady}/\sigma \approx 0.07$.

### Moment-duration scaling relations of SSEs

We simulate single-rupture models by prescribing different values of initial shear stress to obtain moment-duration scaling relations of SSEs. The other model parameters are fixed and are the same as those described above, except for a smaller $W/L_c = 100$ to reduce the computational cost and thus allow for a longer simulated fault, $20W$. For the homogeneous shear stress model, the stress drop is always lower than the steady stress drop, $\Delta\tau^{steady}$, which only results in self-arresting ruptures. For the linearly decaying shear stress model, the initial shear stress is largest near the nucleation zone and linearly decreases to zero at the other side of the fault. To smoothly nucleate unilateral ruptures,

we prescribe a minimum nucleation length, $0.1W$, with higher slip rates. For each rupture model, we determine the rupture length, $L$, by the final position of the rupture tip, and we estimate the SSE duration, $T$, based on a slip rate threshold, $0.1V_c$. Note that the SSE duration is slightly longer than the rupture time by a rise time. As the prescribed initial shear stress increases, the rupture length, $L$, moment, $M_0$, and duration, $T$, of the SSEs increases accordingly. In the homogeneous shear stress model, $L$ and $M_0$ increase toward infinity as stress drop asymptotically approaches $\Delta\tau^{steady}$.

For long ruptures, the moment is $M_0 - \Delta\tau W^2 L$, where $L$ is the rupture length and $\Delta\tau$ and $v_r$ are the average stress drop and rupture speed, respectively. The rupture duration is $T \approx L/v_r$. The theory and numerical simulations predict two characteristic quantities for steady SSEs in above equations: $\Delta\tau^{steady} \approx 0.01\sigma$ and $v_r^{steady} \approx 50V_c\mu/\sigma$. Therefore, the moment and duration can be normalized as

$$\frac{M_0}{\Delta\tau^{steady}W^3} \sim \frac{\Delta\tau}{\Delta\tau^{steady}} \cdot \frac{L}{W}$$
$$\frac{T}{W/v_r^{steady}} \sim \frac{v_r^{steady}}{v_r} \cdot \frac{L}{W}. \tag{38}$$

In the numerical simulations, $L/W$, $\Delta\tau/\Delta\tau^{steady}$, and $v_r/v_r^{steady}$ are calculated. Defining $\Delta\tau/\Delta\tau^{steady} \propto (L/W)^\alpha$ and $v_r/v_r^{steady} \propto (L/W)^\beta$ leads to

$$\frac{M_0}{\Delta\tau^{steady}W^3} \sim \left(\frac{T}{W/v_r^{steady}}\right)^{\frac{1+\alpha}{1-\beta}}. \tag{39}$$

In the homogeneous shear stress model, if $\Delta\tau > \Delta\tau^{steady}$, ruptures steadily propagate through the entire fault with $\Delta\tau$ and $v_r$ independent of rupture length $L$, that is $\alpha = 0$ and $\beta = 0$. If $\Delta\tau < \Delta\tau^{steady}$, self-arresting ruptures decelerate and gradually stop for various values of $\Delta\tau$, which roughly results in $\alpha = 0.25$ and $\beta = -0.25$ (Fig. S3). Therefore, both steady and self-arresting ruptures in the homogeneous stress model produce a linear moment-duration scaling relation. However, in the linearly decaying shear stress model, the simulated models result in $\alpha = 0.5$ and $\beta = 0.5$ (Fig. S3), which leads to a cubic scaling relation.

Because $\Delta\tau \propto \sigma$ and $v_r \propto 1/\sigma$ (equations (21), (34), and (23)), the dimensional analysis of equation (38) shows that $\alpha$ and $\beta$ are independent of $\sigma$, $M_0 \propto \Delta\tau^{steady} \propto \sigma$, and $T \propto \frac{1}{v_r^{steady}} \propto \sigma$, which is also validated by numerical simulations that are not shown here. Therefore, as $\sigma$ systematically varies, the cubic scaling curve between $M_0$ and $T$ moves diagonally in the $M_0 - T$ space (Fig. 4b).

In addition, a similar dimensional analysis shows that $M_0$ is independent of $V_c$ and $T - 1/V_c$. As $V_c$ systematically varies, the scaling curve moves vertically in the $M_0 - T$ space, that is $T \propto 1/v_r \propto 1/V_c$, which can reconcile the separation between the cubic scaling of SSEs and earthquakes.

## Data availability
The numerical data have been deposited in the Zenodo database under accession code 10.5281/zenodo.7228123 [https://doi.org/10.5281/zenodo.7228123]. The theoretical data are presented in the Methods. Other data are previously published and available in the references cited in the figure captions.

## Code availability
The open source softwares SEM2DPACK and QDYN used in the fully dynamic and quasi-dynamic rupture simulations are available at https://github.com/jpampuero/sem2dpack and https://github.com/ydluo/qdyn, respectively.

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

## Acknowledgements

This manuscript benefited from discussions with M.P.A. van den Ende, Quentin Bletery and Jean-Mathieu Nocquet. We thank Frédéric Cappa and Allan Rubin for their constructive comments in reviewing the early version of this manuscript. This work was supported by the French government through the Investments in the Future project UCAJEDI (ANR 15 IDEX 01) managed by the French National Research Agency (ANR). H.W. was also partially supported by the National Natural Science Foundation of China. We are grateful to the "MéSOceNtre SIGAMM" at Observatoire de la Côte d'Azur and to Université Côte d'Azur's OPAL infrastructure and Center for High-Performance Computing for providing resources and support.

## Author contributions

H.W. conceived this study, carried out the numerical experiments and the theoretical derivations. J.-P.A. contributed the ideas to compute the average slip rate of the observed SSEs, the recurrence time of earthquakes, and the moment-duration scaling resulting from the model. H.W. wrote the paper, J.-P.A. edited it. Both H.W. and J.-P.A. contributed to the revision of the paper.

## Competing interests

The authors declare that they have no competing interests.
