## [Peer Review File · Nature Communications]

REVIEWER COMMENTS

Reviewer #1 (Remarks to the Author):

This manuscript presents an explanation for the observation of two different scaling (linear vs. cubic) of the moment-duration relation of slow slip events. Using numerical and theoretical tools, the author demonstrates that two different types of fault heterogeneities may lead to these scaling at the local vs. global level. The paper further shows a possible link between the cycles of slow slip events and the supercycles of earthquakes. Both aspects are of fundamental importance for earthquake research and have wide implications for seismology.

The presented work is thorough and original. I have a few suggestion that I think would be valuable for the manuscript.

Main comments:

1) One major comment relates to the link to previous work on the concept of energy balance for earthquake rupture propagation. Specifically Svetlizky et al., 2017 <https://doi.org/10.1103/PhysRevLett.118.125501> [S2017] for slow and sub-shear speeds, and Kammer et al., 2018 <https://doi.org/10.1126/sciadv.aat5622> [K2018] for supershear propagation. These works demonstrated that the G_c/G_0 ratio governs propagation speed including slow ruptures. However, they assumed G_c to be a constant interface properties (based on experimental observations on PMMA). Due to this assumption, slow slip events are theoretically possible but not common because the requirements on the energy release rate G_0 to maintain a small G_c/G_0 required for slow speed are very restrictive. In the present work by Huihui Weng, he elegantly introduces a non-monotonic rate-dependence into G_c , which enables a mechanism that allows for low G_c/G_0 values and hence can lead to sustained slow slip events. Nevertheless, I believe it would be appropriate to present the current work in the context of this prior work. This touches various points in the manuscript, I point to a few:

- l. 27: "laboratory experiments", [S2017] and [K2018] demonstrated already the range of rupture speeds across the full spectrum up to supershear.

- l. 43-44: "In particular, substituting $v_r = 0$ into equation (1) yields the energy balance condition for a steady-state rupture: $G_c(v_r) = G_0(v_r)$." This was experimentally demonstrated by [S2017] and [K2018].

- l.60 and 80: both state that (1) is applicable. Are both cases non-steady state (this part is not entirely clear, see comment 3)? If either is related to state-state, then [S2017] provides experimental confirmation for it.

2) Throughout the paper, I remain confused about the energy release rate. For a dynamic earthquake (in an unbound domain), the energy release rate depends on the rupture velocity $G_0(v_r)$. However, this effect can be neglected for slow slip event. It remains unclear if this aspect was considered for the dynamic events, as various statements appear to be contradictory.

- l. 43-44: "steady-state rupture $G_c(v_r) = G_0(v_r)$ ", here the effect is considered.

- l. 60-62: "In addition, G_c increases with v_r while G_0 is a prescribed parameter independent of v_r ", here it says it is independent.

- l. 68-70: α_s is applied to the speed ratio v_r/v_s , but it does not state on what quantity this is done. The energy release rate? This paragraph describes the dependence of rupture speed on stress drop, but Method A6 referenced for the α_s term for peak slip rate.

- l. 159 + method A5: $G_0 = \lambda \Delta \tau^2 W / \mu$. Why is it independent of the rupture length? Is it because the material depth is limited and remains constant along the propagation direction, hence, it is a strip configuration? If so, please add a short reasoning for it.

- Eq. (29): the energy release rate appears to be static. Is that correct? Is this equation only applied to SSE?

3) For section "General mechanics for steady SSEs and earthquakes", the reader needs a bit more guidance. For instance, the difference between the paragraph l.48-62 and the paragraph l.63-82, remains unclear to me. Both end with the statement that Eq. (1) is

demonstrated to be applicable. What is the difference? It would be advisable to tell the reader more explicitly about the purpose of the various steps during the derivation of the conclusion.

Minor comments:

- l. 13-16 (abstract): "The complete spectrum of rupture speeds, from arbitrarily slow speeds up to the S-wave speed, is quantitatively controlled by the ratio of fracture energy to energy release rate, which depends on the frictional behaviour on the fault." This is technically not incorrect, but could be misunderstood. The energy release rate does not only depend on the frictional behavior of the fault (which I understand as the "friction law") but also on the stress state (as you also investigate in your manuscript). I would suggest to rephrase this sentence.
- l. 43-44: "Both G_c and G_0 are functions of rupture speed and depend on the friction behaviour of the fault v_r " same as previous comment.
- l. 83: " $G_c = G_0$ is controlled by the specific frictional behaviour of the fault" same as previous comment.
- I may have missed where it was stated, but it is my understanding that this is anti-plane slip, i.e., slip is perpendicular to rupture speed. Is this correct? Is it stated?
- l. 55: "The rupture speed increases monotonically with the stress drop ...". This is the consequence of the energy release rate in a homogeneous system.
- l. 159: "The energy release rate on a long fault is $G_0 = \dots$ " this assumes uniform stress. Elsewhere in the paper, a heterogeneous setup is mentioned. Please clarify.
- l. 161: " $B = 1.18$ ". This means that G_c (l. 157) has only a slight but inverse dependence on normal stress. I find this surprising. What is the reasoning for this assumption?
- l. 223: the mentioned gap is unclear in Fig. 2d. Is it the gray area? The values reported in the text are absolute and the figure has normalized values.
- Fig. 3 inset: could use a label for the x-axis.
- l. 224. "The numerical models show ... (Figure 2a)". This sentence refers to rupture speed but Fig. 2a shows slip rates. Both are related, as shown in the paper, but this should be reformulated.
- A1: uses D as average slip, whereas D was final slip in the main text.
- after eq. (7): " $F()$ " should be " $f()$ "
- after Eq. (10): " $G_0(v_r) > 0$ " what is the reasoning for this?
- after Eq. (11): " $F(0, v_r) = 1$ " should be " $f(0, v_r) = 1$ ".
- l. 295: "the loading due to the plate convergence during rupture propagation is not considered". I do not understand. Please clarify.

Reviewer #2 (Remarks to the Author):

The presented manuscript builds on previous theoretical work by the author (Weng and Ampuero, JGR 2019) considering elongated frictional ruptures (earthquake ruptures) in the framework of dynamic fracture theory. In the present manuscript, the author aims to use the framework of linear elastic dynamic fracture theory to describe ruptures with a broad range of rupture speeds, from aseismic slow-slip ruptures to large dynamic earthquake ruptures. Essentially the author argues that this full range of rupture speed behavior can be explained in terms of the balance of fracture energy (the energy that must be consumed to advance the crack) and the energy release rate (the mechanical energy supplied to the crack tip).

The theory presented in Weng and Ampuero (2019) for geometrically-restricted 3D cracks was shown to work well for describing simulated frictional ruptures using slip-weakening friction, which has been well-established to be mathematically equivalent to cohesive zone models of linear elastic fracture mechanics (LEFM). In other words, the friction law used in the previous study by definition is compatible with dynamic fracture theory. Similarly, standard logarithmic rate-and-state friction laws, such as that used in the presented work, have been shown to resemble linear slip weakening (e.g. Okubo, 1989; Cocco and Bizzarri, 2002) and hence are mathematically compatible with LEFM.

Two general questions arose while reading the current manuscript:

1) How generally appropriate is the strict analogy between linear elastic dynamic fracture theory and frictional ruptures (i.e. how general are the results of this work to more realistic constitutive models of fault shear resistance), given the mounting evidence suggesting that the key underlying assumptions to this mathematical analogy are violated when considering processes like enhanced dynamic weakening at seismic slip rates (as widely documented in laboratory experiments and considered to be predominant during earthquake ruptures) as well as damage generation (more details below)?

2) Assuming that LEFM can generally be used to describe all types of frictional ruptures (slow-slip events to dynamic earthquake ruptures), what are the specific novel ideas presented in this work compared to prevailing views applying fracture mechanics for earthquake mechanics?

For readers who are not familiar with dynamic fracture theory, the current manuscript may seem quite technical. Following the arguments in the main text required continuous referral to the supplement (and the author's previous work, Weng and Ampuero JGR 2019) in order to follow along. For those who are more familiar with dynamic fracture theory and related literature, the key take-aways from the presented work seem to be a review of existing theoretical concepts and standard views of rupture growth. For example, it is well established for crack propagation in linear elastic theory that the balance between fracture energy and energy release rate controls rupture speed and crack growth, and an increase in fracture energy ahead of the crack tip is needed to balance the growing energy release in order to maintain a constant rupture speed (as described by many prior studies and discussed in detail in Freund, 1990 Dynamic Fracture Mechanics, Cambridge Monographs).

My understanding of the main arguments presented in the abstract, namely that a complete spectrum of rupture speeds is realizable for frictional rupture in nature and that the full range of earthquake and slow-slip transient behavior are fully described by traditional fracture mechanics, do not clearly appear to be supported the presented work. However, there are some interesting points that could be potentially explored/discussed in detail, such as why existing observations don't appear to strongly suggest a full continuum of realized rupture speeds, but rather two groups of slow-slip or dynamic events, and if the inferred global linear moment-duration scaling of slow slip events suggests a systematic variation in stress drops among regional populations of slow slip events.

I expand on the two questions below and include additional comments/questions afterwards.

%%%%%%%%%%
%%%%%%%%%%

1) How appropriate is the mathematical analogy between linear elastic dynamic fracture mechanics hold for frictional ruptures, particularly in terms of determining rupture speed?

The presented work relies heavily on the assumption that standard linear elastic dynamic fracture mechanics applies to the broad array of frictional ruptures from slow-slip events to earthquake ruptures. While dynamic fracture theory works well for slip-weakening friction laws that are mathematically identical to cohesive zone laws of LEFM, it isn't clear how valid this analogy is for more realistic fault constitutive laws that include continuous weakening or variable dissipation behind the rupture tip. In other words, the work seems to limit itself to frictional formulations that are comparable to slip-weakening friction, though there is substantial evidence that such formulations may not be complete descriptions of fault friction. It is thus not clear whether the presented modeling results are specific to the chosen friction law and are necessarily relevant to natural slow-slip events and earthquakes.

The mathematical analogy between singular cohesive-zone models of shear cracks and frictional ruptures depends on two key assumptions (Palmer and Rice, 1973):

- (i) The breakdown of shear resistance during rupture is concentrated in a small region near the rupture front (i.e., small-scale yielding); and,
- (ii) A constant residual stress level exists throughout the ruptured region during sliding, such that the sliding stress level is comparable with the final stress level after rupture (allowing for this residual stress to be subtracted to draw analogy to cohesive zone models of mode I cracks).

In particular, the clear relationship between rupture speed and the fracture energy of linear elastic fracture mechanics is only valid under these assumptions.

This work explores rate-and-state friction including some rate-strengthening at high slip rates, "as observed in laboratory experiments" (lines 49-51), which is generally consistent with slip-weakening friction. The work does mention that enhanced dynamic weakening can occur at high slip rates, which in fact is widely documented in laboratory experiments (e.g. Di Toro et al., Nature 2011), however there is no discussion of how this alters the underlying assumption that traditional dynamic fracture theory is valid.

Recent numerical studies have demonstrated that both of these key assumptions become invalid with the inclusion of thermo-hydro-mechanical processes, such as the thermal pressurization of pore fluids, which has been suggested to be prevalent for nature earthquakes of varying size (Abercrombie and Rice, JGR 2006; Viesca and Garagash, Nature Geo 2015; Lambert and Lapusta, Solid Earth 2020). Enhanced dynamic weakening can result in continued or variable weakening well behind the rupture front and the associated local breakdown energy (thought to be the frictional equivalent to fracture energy) is rupture-dependent, i.e. not a fixed property of the interface. In such case, the original analogy to traditional fracture theory is no longer obviously justified and the relationship between breakdown energy and rupture speed is not clear. Moreover, the relationship between the energy balance of frictional ruptures and the standard energy budget based on dynamic fracture theory depends on the rupture style, for example self-healing pulse-like ruptures are not well-described by the standard energy budget (Heaton, PEPI 1990; Viesca and Garagash, Nature Geo 2015; Lambert et al., Nature 2021).

Note that other mechanisms for variable breakdown energy during dynamic rupture, such as production of damage and off-fault inelastic deformation (e.g. Andrews, JGR 2005; Okubo et al., JGR 2019) would likely also violate these assumptions as the relevant dissipation for the dynamic is not constrained to the rupture tip.

Overall, how does the presented work account for or comment on the accumulating body of work that question the underlying assumptions for the analogy between traditional dynamic fracture theory and frictional ruptures?

On a related note, does the presented modeling reproduce the well-known inferred increase in average breakdown energy with average slip and/or rupture size of earthquakes, as determined for natural earthquakes and in laboratory experiments (Abercrombie and Rice, JGR 2005; Rice, JGR 2006; Cocco and Tinti, 2008; Viesca and Garagash, 2015; Nielsen et al., 2016; Brantut and Viesca, 2018; Selvadurai, JGR 2019). Such inferences have typically been used to suggest that thermal pressurization or some other dissipation (such as along off-fault damage) are prevalent across earthquakes of a broad range of sizes, unless some special heterogeneity exists along the fault.

%%%%%%%%%%
%%%%%%%%%%

2) Assuming that linear elastic dynamic fracture theory can be appropriately applied to frictional ruptures in general, what novel insight is supported by the presented modeling?

The balance between fracture energy and energy release rate in determining rupture speed is a well-established aspect of dynamic fracture theory and fracture mechanics has been

widely applied to study aspects of earthquake ruptures (e.g. Palmer & Rice, 1973; Madariaga, 1976; Kostrov and Das, 1998; Kanamori and Brodsky, 2004; Rubin and Ampuero, 2005). The proposal that fracture mechanics works well for describing ruptures governed by rate-and-state friction is not really novel, and the generality of its strict application is questionable as noted above.

Key point #1 from the abstract (lines 13-15) is that a "complete spectrum of rupture speeds Is quantitatively controlled by the ratio of fracture energy to energy release rate, which depends on the frictional behavior on the fault."

This is a well-established aspect of dynamic fracture theory but would not clearly be evident when small-scale yielding is violated, as mentioned earlier. Moreover, the energy release rate may not just be a function of frictional behavior along a fault, but weakening or strengthening can depend on other mechanisms such as damage generation or enhanced dynamic weakening such as from thermal pressurization. In the case of enhanced weakening, large dynamic stress changes due to efficient thermal pressurization would depend on the dynamics of the individual rupture and increase the energy release rate and may not be strictly related to the static stress changes: (Heaton, PEPI 1990; Lambert et al. Nature 2021).

Key point #2 from the abstract (line 16-18) addresses the difference in linear versus cubic moment-duration scaling of slow-slip events, where heterogeneity in shear stress produces cubic scaling while heterogeneity in effective normal stress produces linear scaling.

The presented arguments for the cubic scaling for ruptures with heterogeneous shear stress but constant rupture speed were not particularly clear in this work. In essence, it would seem that this argument really follows the results of Dal Zilio et al., GRL 2020, who demonstrated that rupture speed can vary substantially during slow-slip events due to heterogeneous shear stress conditions in finite ruptures and accounting for this variable rupture speed reproduces the cubic moment-durations scaling. It is not clear how a similar result arises from scaling arguments with a constant rupture speed.

There were two points that seemed potentially interesting, with the potentially major caveat that the theoretical discussion based on dynamic fracture theory may not strictly apply:

1. Spectrum of rupture speeds:

The proposed results from dynamic fracture theory predict a spectrum of rupture speeds is possible depending on critical strengthening velocity. However, a complete spectrum of rupture speeds may not actually be observed, and hence realized.

Figure 2a demonstrates a potential continuum of rupture speeds in the modeling, however existing inferences from natural events in Figure 1b don't actually seem to demonstrate a continuum. How evident is it that there is a "continuum of rupture speeds" for slow-slip to dynamic earthquake ruptures? It would appear that all of the earthquake rupture speeds are concentrated in one group (dynamic events) and then slow-slip events are clustered together as well, with the large gap of several orders of magnitude in slip rate and rupture speed being somewhat filled with a few inferences from Izu-Bonin SSEs and some lab experiments. Is it possible that there is not actually a continuum of plausible rupture speeds but instead two favorable regimes?

Two potential avenues are suggested by this work: 1) More observations are needed to fill in this gap; 2) the critical velocity is low enough that regions are separated into two different stability regimes

□ Can the author comment more on the relationship between this critical velocity and different physical mechanisms (e.g. strength of dilatancy vs thermal pressurization) to suggest why there may not be a realized continuum of rupture speeds but ruptures are clustered as either slow-slip or dynamic events? On lines 217-218 the author claims that this critical slip rate can be "constrained" with theoretical arguments, however this range is

essentially the full range of slip rates from around typical tectonic plate rates to near seismic slip values. Can a more informative constraint be made with these considerations that a continuum may not be realized?

- The friction law used includes some rate-strengthening above a critical value, presumably consistent with models of gouge dilatancy during the initial acceleration of slip? Do any laboratory experiments show continued strong rate-strengthening at seismic slip rates?
- How would your results differ if you included a second critical velocity with a transition in weakening behavior, such as strong rate weakening friction of flash heating or thermal pressurization (Rice, JGR 2006)?

2. Moment-duration scaling of slow slip events

The main argument seems to be that previous work (e.g. Dal Zilio et al., GRL 2020) has demonstrated that cubic scaling is possible when accounting for spatially variable stress changes and rupture speeds during slow-slip events, however if one combines multiple populations with cubic scaling then the relation can appear linear if stress drop varies among populations.

- Comparing regional slow-slip population moment-duration scaling in addition to global scaling seems to be a potentially insightful practice. It would seem that the argument here for an overall linear global trend may suggest a systematic difference in stress drop among slow-slip event populations. Do observations from natural slow-slip events support this, i.e. do some slow-slip event populations have systematically lower or higher average stress drops consistent with this shift to make a linear scaling?

Please find below additional comments and questions:

%%%%%%%%%%
%%%%%%%%%%

Note that the line numbers are missing from parts of the pdf which makes it challenging to specifically reference lines in the text.

Figure 3b: It isn't clear what the x and y axes are in the inset.

Lines 92-93: "weakening controlled by thermal pressurization ... is slip-dependent rather than rate-dependent." This is not true. Recent numerical studies (Lambert and Lapusta, Solid Earth 2020) including thermal pressurization have demonstrated that weakening behavior due to thermal pressurization is not a unique function of slip but depends on the history of sliding rate and dynamic stress changes throughout individual ruptures. The same point along a fault can experience different weakening behavior with comparable amounts of slip in different ruptures due to differences in the slip rate function and dynamic stress changes. This in turn means that local breakdown energy is 1) rupture-dependent and 2) does not strictly increase or decrease with local slip. The pure slip-dependence of thermal pressurization in early theory assumes a constant sliding rate, or a consistent slip rate function at all points along a fault, which is not the case in finite ruptures.

Line 96: Why is D necessarily bounded by the interseismic slip deficit for individual points during the rupture? It is well demonstrated in dynamic rupture models that points throughout ruptures can exhibit a dynamic overshoot and slip more than would be expected in a quasi-static calculation based on the accumulated static stress to drop. A point can overshoot slip and then account for this with extra locking during the subsequent interseismic period.

Lines 167-171. The inferred low-stress operation of mature faults is with respect to typical Byerlee values of friction (~ 0.6) consistent with steady-state friction coefficients measured in the lab (perhaps consistent with the shear stress conditions for nucleation), not the peak friction, which could in fact be much higher. Note that the peak shear stress depends on how well-healed the fault is through the state variable as well as the slip rate during rupture. Given the mild logarithmic rate-weakening of standard rate-and-state friction (as used in this work), the average stress state along the rupture area is within one static stress drop of this quasi-static strength for nucleation, so such models (including those presented in this work) would need either low quasi-static reference friction or the fault

would need to have low effective normal stress, such as through substantial fluid overpressure, to be compatible with the inferred low effective friction conditions.

Equation (16) the meaning of L_c is not defined – process zone size

Rupture time is computed for the numerical models by the slip rate criterion $V = 10V_i$, so this depends on the choice of V_i ? Have you investigated the sensitivity of timing to choice of threshold? Particularly for slow-slip events where slip rates and slip acceleration can vary substantially?

Doesn't Equation (17) assume a uniformly prestressed crack or constant stress drop? Do you compute energy release rate directly in your simulations, or use this equation?

Here, the energy release rate is considered a function of static stress drop once again in the analogy to cracks where a traction-free surface occurs behind the crack tip and essentially all break down or dissipation occurs at the rupture tip. In other words, the static stress drop is a reasonable measure of the total stress change during the dynamic process. For frictional ruptures, dissipation can occur throughout the entire rupture and the energy release rate would be related to the difference between the total strain energy change and the energy that is dissipated, along the fault or in creating new fractures off-fault such as with damage. So the energy release rate is not strictly related to the stress drop but would be a more complicated consideration of the dynamic stress changes. For example, the static stress drop could vastly underestimate the dynamic stress changes in the case of a self-healing pulse with a substantial stress undershoot (e.g. Lambert et al., Nature 2021).

Equation (18) assumes that the final level of shear stress is equal to the minimum shear resistance during sliding, which would not be the case for an overshoot or undershoot, for example in the case of a self-healing pulse (Heaton, PEPI 1990, Viesca and Garagash, Nature Geosci 2016, Lambert et al., Nature 2021). For rate-and-state friction with mild slip rate dependence of the shear resistance, the level of sliding resistance and final stress may be comparable, with mild overshoot. However this need not be the case for enhanced dynamic weakening, as may be expected to dominate during earthquake ruptures.

Before equation (26), $T_r - T_f$ is not the traditional definition of overshoot, which is rather the stress adjustment due to waves after slip has arrested. This quantity $T_r - T_f$ would just be continued weakening during slip within the rupture surface. From dynamic fracture theory It isn't clear how relevant this would be for driving the rupture tip.

How is stress drop being estimated in these models? Is it uniform? The spatial average? Is this energy release rate assuming uniform stress drop throughout the rupture? This would certainly not be the case natural ruptures, which is even evidenced from finite-fault inversions (e.g. finite-fault inversions of large earthquakes in Ye et al., JGR 2016ab). Does this critical stress drop for steady propagation also assume that stress drop is uniform within the rupture area?

Line 364: In the homogeneous model where the stress drop $>$ critical value, ruptures steadily propagate throughout the entire fault. Does this mean that the ruptures don't naturally arrest but are forcibly arrested by the model geometry, i.e. the VS boundaries? How do the average rupture properties – duration, stress drop, rupture speed – depend on the properties of the VS boundaries? If one wants to compare scaling relations among aspects of the full finite rupture then accounting for these regions of rupture propagation and arrest would seem to be important, particularly if the VS boundaries are not very strong and propagation continues substantially into these regions (?).

Equation (31): Presumably the factor of $(1-\nu)$ in the "strike-slip" term should be related to mode II rupture (vs μ for mode III), which for a megathrust fault would be dip-slip.

Reviewer #3 (Remarks to the Author):

Review for “Integrated rupture mechanics for slow slip events and earthquakes” by Huihui Weng

The author has established a theoretical framework based on fracture energy balance at the rupture tips in order to integrate the global observations of slow slip and earthquakes in a unified physical model. The author has provided a comprehensive mathematical derivation in the methodology, conducted a lot of numerical simulations, and analysed the global observations of slow slip and earthquakes in various stress environments. The work gives a physical explanation of the wide range of rupture speed, which might be helpful for us to understand the unified mechanism underlying tectonic faulting system.

However, I have several concerns for publishing it in current state. First of all, the work evolves lots of specific mathematical in the main text and doesn't fit well into a letter journal. I need to refer to the method part quite a lot when reading which took me lots of for the mathematical analysis in the main text. I feel it would be appropriate for a longer-type paper.

Secondly, integrating the source characteristics of slow slip and earthquakes in the framework of rate-state frictional faults in numerical simulations is not surprisingly new. Previous work from laboratory experiments and numerical simulations has proved that well (e.g. Leeman et al, 2016; Barbot 2019).

Thirdly, the writing of the paper is not clear enough in some way such that I had a hard time to follow it, even through I find myself quite familiar with SSEs. I could image readers from other backgrounds suffering from it. Therefore I would suggest either reformulating the paper or considering another long-type journal, like JGR.

Major comments:

The word 'steady-rupture' appeared lots of times but I don't understand. Earthquakes start from acceleration in slip and then decrease to die out. The process is sensitive to not just structural heterogeneities but also various physics such as thermal weakening and pore fluid changes as also mentioned in the manuscript. I don't understand how does 'steady' explains the process. Maybe the author means 'sustain-rupture'? Either change the word or explaining it properly in the context.

Some quantitate in the research are not defined clearly. For example, the definition of stress drop could be various depending on the averaging methods (refer to 3.1-3.3 in Perry et al. 2021). There is quite a variation in the 2D forward model and larger variations should be expected in 3D model. Critical slip rate V_c is an important parameter in the analysis, however, it is not clear how V_c is determined in the model and how it is supported by experiments. Why don't use the more standard RSF format? Will it result in different tuning parameters?

The 2.5D model uses a uniform velocity model (at least I guess so since it is not mentioned of the velocity model in A4), which will ignore the effects of material rigidity. It has been shown in a lot of studies that 3D structure will affect the rupture speed, especially for shallow tsunami earthquakes (e.g. Lay et al. 2012). The source depths of SSEs also span a wide range of temperatures and materials which play an important role. There is, however, only one sentence on line 251 discussing this effect.

The discussion of the moment-duration scaling part is interesting. The author attributes the “debated scaling behaviors can be attributed to different types of fault heterogeneities: heterogeneity of shear stress can produce a cubic scaling, whereas heterogeneity of effective normal stress produces a linear scaling.” What implication on tectonic stress environments can be learned from these heterogeneities? What kind of volumetric tectonic loading generates only shear stress or normal stress heterogeneity?

The introduction is not quite related to the framework of the paper. I would suggest reshaping it a bit to explain what is the correct research gap between slow and fast earthquakes and why is the linear or cubic scaling between seismic and aseismic events important. Which research questions does this work address?

Minor comments:

line 93: the expression is not correct.

line 96: I don't think it is a valid assumption, considering only partial ruptures and the less well-constrained slip deficit inversion. A recent example could be 2011 Tohoku-oki event where shallow less coupled fault ruptured with maximum slip.

line 104: I don't think it is correct. Segall and Bradley shows that the repeated SSEs produce a stress concentration at the top of the ETS zone which causes thermal pressurization that is sufficient to allow rupture to accelerate to inertial limits. I find, however, this is not consistent with the statement that "this steady SSE could transition to a non-steady earthquake and accelerate toward the S-wave speed". Additionally, the model here where G_0 is constant and not changeable during cycles?

line 108-109: I don't understand what is need to be confirmed.

line 141: effective normal stress should be $\bar{\sigma}$

line 142: I don't think it is a stress drop, might be slip? The formula might be wrong since a parenthesis is missing for $\pi\mu$

lines 150-153: this part is not new and not well related to the paper

Line 170: I don't think it is correct. The author compares $\mu_r + 0.5 \times \text{average_stress_drop} / \text{normal_stress}$, which equals to $\tau_0 / \text{normal_stress}$, and since τ_0 is always smaller than peak τ (τ_p) in RSF, the comparison doesn't have anything to do with static fault stress (τ_p is not τ_{static}). Additionally, I don't think this derivation of the relation between fault average strength and static strength is novel in this study.

line 181: 'discussions' -> 'discussion'

line 221: 'and' to 'whereas'

lines 222-242: this paragraph is based on V_c but there is no physical meaning of V_c yet. The cut-off velocity in the specific RSF used in this paper is not a standard format. Why not use the standard RSF format?

References:

Perry, S. M., Lambert, V., & Lapusta, N. (2020). Nearly magnitude-invariant stress drops in simulated crack-like earthquake sequences on rate-and-state faults with thermal pressurization of pore fluids. *Journal of Geophysical Research: Solid Earth*, 125, e2019JB018597. <https://doi.org/10.1029/2019JB018597>

Lay, T., H. Kanamori, C. J. Ammon, K. D. Koper, A. R. Hutko, L. Ye, H. Yue and T. M. Rushing (2012). "Depth-varying rupture properties of subduction zone megathrust faults." *Journal of Geophysical Research: Solid Earth* 117(B04311): B04311.

Leeman, J. R., D. M. Saffer, M. M. Scuderi and C. Marone (2016). "Laboratory observations of slow earthquakes and the spectrum of tectonic fault slip modes." *Nat Commun*.

Barbot, S. (2019). "Slow-slip, slow earthquakes, period-two cycles, full and partial ruptures, and deterministic chaos in a single asperity fault." *Tectonophysics*.

Comments from reviewer #1:

1. One major comment relates to the link to previous work on the concept of energy balance for earthquake rupture propagation. Specifically Svetlizky et al., 2017 <https://doi.org/10.1103/PhysRevLett.118.125501> [S2017] for slow and sub-shear speeds, and Kammer et al., 2018 <https://doi.org/10.1126/sciadv.aat5622> [K2018] for supershear propagation. These works demonstrated that the G_c/G_0 ratio governs propagation speed including slow ruptures. However, they assumed G_c to be a constant interface properties (based on experimental observations on PMMA). Due to this assumption, slow slip events are theoretically possible but not common because the requirements on the energy release rate G_0 to maintain a small G_c/G_0 required for slow speed are very restrictive. In the present work by Huihui Weng, he elegantly introduces a non-monotonic rate-dependence into G_c , which enables a mechanism that allows for low G_c/G_0 values and hence can lead to sustained slow slip events. Nevertheless, I believe it would be appropriate to present the current work in the context of this prior work.

A. There is a fundamental difference between our work and the previous body of work found in the literature: our work accounts for the finite rupture width W . This feature is essential to study large earthquakes and Slow Slip Events, because their ruptures are longer than wide, but is missing in previous theoretical and laboratory studies. As shown in Weng and Ampuero (2019), the effect of W changes drastically the scaling of energy release and thus the rupture dynamics. In particular, the energy release rate of steady elongated ruptures, $G_0 = \Delta\tau^2 W / \pi\mu$, is independent of rupture distance and rupture speed, thus a steady state (satisfying $G_0 = G_c$) can be realized naturally even under a G_c that does not scale with rupture size. Classical theories and experiments that do not include a finite W are not conclusive for the processes studied in this paper, namely large earthquakes and SSEs. For this reason, our study did not rely much on works such as those cited by the reviewer.

In the revised manuscript, we made an effort to better highlight the originality of our work and its relation to previous relevant work:

Numerical and theoretical studies^{53,54} demonstrated that **the energy balance governing rupture propagation on long faults with finite width in a 3D elastic medium depends on the rupture width rather than on the rupture length. In particular, Weng and Ampuero⁵³ developed a 3D theory of large ruptures that accounts for their finite rupture width W . Their theory yields a rupture tip equation of motion of the following form** (Methods A2):

$$F\left(\frac{G_c}{G_0}\right) = M(v_r) \dot{v}_r \quad (1)$$

where G_c/G_0 is the ratio between the fracture energy and the energy release rate of steady-state ruptures, \dot{v}_r is the rupture acceleration, the time derivative of rupture speed v_r , and F and M are known universal functions. **Here, the energy release rate of steady-state ruptures is exactly equal to the static energy release rate and, in contrast with the classical 2D fracture mechanics theory, it is independent of the rupture length⁵³:**

$G_0 = \Delta\tau^2 W / \pi\mu$, where $\Delta\tau$ is the static stress drop and μ is the shear modulus. These features of G_0 arise because the finite fault width W turns the rupture into a slip pulse with length smaller than W , and are similar to those of a rupture in a 2D strip of thickness W (ref^{55,56}). The finite W , an essential ingredient for the study of large earthquakes and SSEs, is missing in previous experimental and theoretical studies of slow and fast ruptures^{43,44}.

...

Further linear stability analysis (Methods A2) shows that the general stability condition for steady-state ruptures can be written as

$$\frac{d(G_c/G_0)}{dv_r} > 0$$

The opposite condition, $d(G_c/G_0)/dv_r \leq 0$, predicts **non-steady ruptures** that either accelerate ($G_c/G_0 < 1$) or decelerate ($G_c/G_0 > 1$). **Generally, equation 1 predicts that there are two types of ruptures on long faults with finite width regardless of the rupture speed: steady and non-steady ruptures. The predicted features of non-steady ruptures with speeds typical of regular earthquakes (close to S-wave speed) have been numerically validated in a previous study⁵³, whereas the predicted steady-state ruptures of both fast and slow speeds are validated by numerical simulations here.**

This touches various points in the manuscript, I point to a few:

- l. 27: "laboratory experiments", [S2017] and [K2018] demonstrated already the range of rupture speeds across the full spectrum up to supershear.

A. The papers cited by the reviewer have demonstrated the rupture speed range between $0.1v_s$ and v_p . In addition, ref 22 (cited in our manuscript) has demonstrated continuous rupture speeds down to much lower values, $v_r \ll v_s$. However, as pointed out in our previous answer, these experiments lack an essential ingredient of large earthquakes and SSEs: their finite rupture width. Accordingly, we added references to these two papers in the Introduction and modified the text as follows:

Though laboratory experiments^{22,34,43,44} have suggested a continuum of rupture speeds, **these experiments lack a finite rupture width, an essential ingredient of large slow and fast ruptures on natural faults**, and the general rupture mechanics **controlled by such finite rupture width** is not completely understood.

- l. 43-44: "In particular, substituting $dvr/dt = 0$ into equation (1) yields the energy balance condition for a steady-state rupture: $G_c(vr) = G_0(vr)$." This was experimentally demonstrated by [S2017] and [K2018].

A. Those two papers do not deal with steady-state ruptures with finite rupture width W , thus they cannot serve as an experimental demonstration of our statement. In the revised manuscript, we clarified it in main text:

Though laboratory experiments^{22,34,43,44} have suggested a continuum of rupture speeds, **these**

experiments lack a finite rupture width, an essential ingredient of large slow and fast ruptures on natural faults, and the general rupture mechanics controlled by such finite rupture width is not completely understood.

and in Methods A2:

Some 2D-analogy laboratory experiments^{22,34,43,44} have suggested a continuum of rupture speeds. However, to our knowledge, there are no laboratory experiments yet on long ruptures with zero-slip conditions beyond a finite width W that could have addressed our problem. The conditions for steady-state rupture in existing experiments, without W effect, are radically different. In existing 2D analog experiments, the energy release rate is a function of rupture speed, rupture length and stress drop: $G(v_r, L, \Delta\tau)$. Because this G does not depend on dv_r/dt (a property often described as “the crack tip has no inertia”), without W effect the steady-state condition cannot be obtained by setting $dv_r/dt = 0$ into the equation of motion. Moreover, G_c needs to be proportional to rupture size L to compensate the linear growth of G with L in order to maintain a steady rupture speed v_r . In addition, because the dependency of G on v_r is negligible when $v_r \ll v_s$, very fine tuning of dG_c/dL is required to obtain a variety of steady slow rupture speeds. Therefore, in the absence of the W effect, steady ruptures can occur only under restricted mathematical conditions. Those features of previous theories and experiments are in striking contrast to the results of our modeling accounting for the W effect.

- l.60 and 80: both state that (1) is applicable. Are both cases non-steady state (this part is not entirely clear. see comment 3)?

A. Line 60 relates to the validation of equation (1) for steady ruptures whereas Line 80 relates to the validation of equation (1) for non-steady ruptures. To clarify this difference, we revised the sentence as:

In all the simulated steady models, G_c agrees with G_0 within 3% (Figure S2a), which validates **the steady-state version of equation (1)**.

If either is related to state-state, then [S2017] provides experimental confirmation for it.

A. [S2017] does not provide experimental confirmation of our equation (1), because their experiments do not feature a finite rupture width W (with zero slip conditions beyond it).

2. Throughout the paper, I remain confused about the energy release rate. For a dynamic earthquake (in an unbound domain), the energy release rate depends on the rupture velocity $G_0(v_r)$. However, this effect can be neglected for slow slip event. It remains unclear if this aspect was considered for the dynamic events, as various statements appear to be contradictory.

A. As proven in Weng and Ampuero (2019), the effect of a finite rupture width W changes

drastically the form of the energy release rate and thus the rupture dynamics of elongated ruptures. In particular, the energy release rate of elongated steady-state ruptures, $G_0 = \Delta\tau^2 W / \pi\mu$, does not depend explicitly on rupture speed. We did consider a possible indirect dependency of G_0 on v_r because the static stress drop $\Delta\tau$ may be a function of v_r in some friction laws, such as the regular rate-and-state friction law. If the static stress drop $\Delta\tau$ is independent of v_r , as is the case for the V-shape rate-and-state friction law used for illustration in our work, then G_0 is independent of v_r . We revised the text below equation 1 to make this point clear:

Here, the energy release rate of steady-state ruptures is exactly equal to the static energy release rate and, in contrast with the classical 2D fracture mechanics theory, it is independent of the rupture length⁵³: $G_0 = \Delta\tau^2 W / \pi\mu$, where $\Delta\tau$ is the static stress drop and μ is the shear modulus. These features of G_0 arise because the finite fault width W turns the rupture into a slip pulse with length smaller than W , and are similar to those of a rupture in a 2D strip of thickness W (ref^{55, 56}). The finite W , an essential ingredient for the study of large earthquakes and SSEs, is missing in previous experimental and theoretical studies of slow and fast ruptures^{43,44}.

In general, both G_c and G_0 can be functions of rupture speed v_r and depend on the friction behaviour of the fault⁵⁷. **In particular, under certain rate-dependent friction laws, G_0 depends on v_r via the dependence of static stress drop $\Delta\tau$ on slip rate, thus on v_r .**

- l. 43-44: “steady-state rupture $G_c(v_r) = G_0(v_r)$ ”, here the effect is considered.

A. This is clarified by the following added text:

In general, both G_c and G_0 can be functions of rupture speed v_r and depend on the friction behaviour of the fault⁵⁷. **In particular, under certain rate-dependent friction laws, G_0 depends on v_r via the dependence of static stress drop $\Delta\tau$ on slip rate, thus on v_r .**

- l. 60-62: “In addition, G_c increases with vr while G_0 is a prescribed parameter independent of vr ”, here it says it is independent.

A. This paragraph assumes a specific V-shaped friction law. In that context, the static stress drop $\Delta\tau$ is independent of v_r , therefore G_0 is independent of v_r . To clarify it, we revised the text as:

In addition, G_c increases with v_r (Figure S2b) while G_0 is independent of v_r **because, with the friction law adopted here, the static stress drop is controlled by the critical slip rate V_c rather than by the peak slip rate** (Methods A5 & Figure S2c).

- l. 68-70: α_s is applied to the speed ratio vr/vs , but it does not state on what quantity this is done. The energy release rate?

A. The original text specified that α_s appears in “the analytical solutions of steady-state ruptures”. The analytical solutions cover all quantities of interest, thus there is no need to list them all.

This paragraph describes the dependence of rupture speed on stress drop, but Method A6 referenced for the α_s term for peak slip rate.

A. The last sentence in Method A6 mentions that combining equations 34 and 29 leads to a relation between stress drop, peak slip rate and rupture speed for both SSEs and earthquakes. To clarify it, we revised the text in Method A6:

Equation 34 describes the relation between peak slip rate and rupture speed. By combining equations 34 and 29, we obtain a relation between stress drop and rupture speed for both SSEs and earthquakes (Figure 2).

- l. 159 + method A5: $G_0 = \lambda \Delta\tau^2 W / \mu$. Why is it independent of the rupture length? Is it because the material depth is limited and remains constant along the propagation direction, hence, it is a strip configuration? If so, please add a short reasoning for it.

A. This was proven mathematically by Weng and Ampuero (2019). The strip configuration is a different problem, but leads to a similar result and can be a useful analogy. To clarify it, we revised the text:

Here, the energy release rate of steady-state ruptures is exactly equal to the static energy release rate and, in contrast with the classical 2D fracture mechanics theory, it is independent of the rupture length⁵³: $G_0 = \Delta\tau^2 W / \pi\mu$, where $\Delta\tau$ is the static stress drop and μ is the shear modulus. These features of G_0 arise because the finite fault width W turns the rupture into a slip pulse with length smaller than W , and are similar to those of a rupture in a 2D strip of thickness W (ref^{55, 56}).

- Eq. (29): the energy release rate appears to be static. Is that correct? Is this equation only applied to SSE?

A. This is clarified in the revised the text:

Here, the energy release rate of steady-state ruptures is exactly equal to the static energy release rate and, in contrast with the classical 2D fracture mechanics theory, it is independent of the rupture length⁵³: $G_0 = \Delta\tau^2 W / \pi\mu$, where $\Delta\tau$ is the static stress drop and μ is the shear modulus.

This is valid for all speeds up to the S-wave speed, which includes both regular sub-shear earthquakes and SSEs.

3. For section “General mechanics for steady SSEs and earthquakes”, the reader needs a bit

more guidance. For instance, the difference between the paragraph 1.48-62 and the paragraph 1.63-82, remains unclear to me. Both end with the statement that Eq. (1) is demonstrated to be applicable. What is the difference? It would be advisable to tell the reader more explicitly about the purpose of the various steps during the derivation of the conclusion.

A. Please refer to our response to question 1.

4. - l. 13-16 (abstract): “The complete spectrum of rupture speeds, from arbitrarily slow speeds up to the S-wave speed, is quantitatively controlled by the ratio of fracture energy to energy release rate, which depends on the frictional behaviour on the fault.” This is technically not incorrect, but could be misunderstood. The energy release rate does not only depend on the frictional behavior of the fault (which I understand as the “friction law”) but also on the stress state (as you also investigate in your manuscript). I would suggest to rephrase this sentence.

- l. 43-44: “Both G_c and G_0 are functions of rupture speed and depend on the friction behaviour of the fault” same as previous comment.

- l. 83: “ $G_c=G_0$ is controlled by the specific frictional behaviour of the fault” same as previous comment.

A. The stress at which a fault operates also depends on the friction properties of the fault. For instance, the stress before each earthquake in a simulation spanning multiple earthquake cycles is a result of the assumptions about friction. In this paper, due to computational limitations, we do not do earthquake cycle simulations, but single-earthquake simulations in which we prescribe the initial stress arbitrarily. For that reason, we prefer not to emphasize initial stress as an independent control parameter.

5. - I may have missed where it was stated, but it is my understanding that this is anti-plane slip, i.e., slip is perpendicular to rupture speed. Is this correct? Is it stated?

A. Yes, that is correct. It is stated in Method A2.

6. - l. 55: “The rupture speed increases monotonically with the stress drop ...”. This is the consequence of the energy release rate in a homogeneous system.

A. The relation between rupture speed and stress drop is not as trivial as the reviewer seems to suggest. For instance, for a steady state rupture on an elongated fault, the relation depends on whether $G_c(v_r)$ increases or decreases with increasing v_r . A detailed explanation follows:

We interpret that the reviewer is referring to the classical 2D fracture mechanics theory, which is very different from our 3D theory with finite rupture width W . In 2D, the energy release rate is a function of stress drop, rupture speed, and rupture distance. Therefore, the 2D equation-of-motion, $G_c(v_r) = G(v_r, L, \Delta\tau)$, predicts that rupture speed increases with stress drop and rupture distance, as the reviewer suggests. However, this is not the case in the 3D equation-of-motion that accounts for the finite fault width W , whose steady-state energy

release rate is $G_0 = \Delta\tau^2 W / \pi\mu$. The energy balance $G_0(\Delta\tau) = G_c(v_r)$ does not explicitly predict that v_r shall increase monotonically with $\Delta\tau$; that depends on whether G_c is a decreasing or increasing function of v_r . Therefore, “the rupture speed increases monotonically with the stress drop” is a statement specific to the friction law adopted in this paper and to the theory with finite rupture width developed here. A more general prediction for the relation between $\Delta\tau$ and v_r has been presented in equations 1 and 2.

7. - l 159: “The energy release rate on a long fault is $G_0 = \dots$ ” this assumes uniform stress. Elsewhere in the paper, a heterogeneous setup is mentioned. Please clarify.

A. We revised this sentence as “**The energy release rate on a long fault assuming a constant stress drop is ...**”.

8. l. 161: “ $B = 1.18$ ”. This means that G_c (l. 157) has only a slight but inverse dependence on normal stress. I find this surprising. What is the reasoning for this assumption?

A. This is not an assumption, but an experimental observation. The value $B=1.18$ is reported by Di Toro et al (ref⁷³) based on the results of their frictional experiments. We think our original text was sufficiently clear about the experimental provenance of these values (“Given $A \approx 3 - 78$, $B \approx 1.18$, and $\mu_p - \mu_r \approx 0.7$ in laboratory experiments (ref 69)”). These authors reasoned that this inverse dependence on normal stress is caused by the inverse dependence of the thermal slip distance on the normal stress $D_{th} = A\sigma^{-1.18}$. This thermal slip distance is taken as the exponential slip-weakening distance in calculations of the fracture energy.

9. - l. 223: the mentioned gap is unclear in Fig. 2d. Is it the gray area? The values reported in the text are absolute and the figure has normalized values.

A. In Line 223, we mentioned the gap of rupture speeds by “there exists an observation gap of rupture speeds in natural faults between 10 –1000 m/s (Figure 1b)” and referred to Figure 1b, not Figure 2d as the reviewer stated. In Figure 1b, the gap of rupture speeds is the white area in between the labeled colored areas.

10. - Fig. 3 inset: could use a label for the x-axis.

A. Revised.

11. - l 224. “The numerical models show ... (Figure 2a).”. This sentence refers to rupture speed but Fig. 2a shows slip rates. Both are related, as shown in the paper, but this should be reformulated.

A. Figure 2a shows the rupture speed, not slip rates. This is correctly stated in the axis label and in the figure caption.

12. - A1: uses D as average slip, whereas D was final slip in the main text.

A. Revised.

13. - after eq. (7): " $F()$ " should be " $f()$ "

A. Thank you. Revised.

14. - after Eq. (10): " $G_0(v_r) > 0$ " what is the reasoning for this?

A. For spontaneous ruptures on long dip-slip faults, the negative G_0 is expected only when $v_r > v_s$. Here we focus on spontaneous ruptures with $v_r < v_s$, therefore $G_0 > 0$ for all cases.

15. - after Eq. (11): " $F(0, v_r) = 1$ " should be " $f(0, v_r) = 1$ ".

A. Revised.

16. - l. 295: "the loading due to the plate convergence during rupture propagation is not considered". I do not understand. Please clarify.

A. We modified the sentence as: "The increase in fault stress due to plate convergence is small during the short time of rupture propagation and thus we ignore it."

Comments from reviewer #2:

1. How generally appropriate is the strict analogy between linear elastic dynamic fracture theory and frictional ruptures (i.e. how general are the results of this work to more realistic constitutive models of fault shear resistance), given the mounting evidence suggesting that the key underlying assumptions to this mathematical analogy are violated when considering processes like enhanced dynamic weakening at seismic slip rates (as widely documented in laboratory experiments and considered to be predominant during earthquake ruptures) as well as damage generation (more details below)?

A. This criticism refers to the applicability of linear elastic dynamic fracture (LEFM) to ruptures with enhanced dynamic weakening and off-fault damage, which has already been validated in previous studies (e.g., Gabriel, et al., 2013; Paglialunga, et al., 2021; Brantut, 2021). We conjecture that reviewer 2 is referring to the “unconventional singularities” found by Brener and Bouchbinder (2021) at intermediate scales (at sufficient distance from the rupture tip) in 2D rupture models with friction laws that transition from weakening to strengthening at increasing slip rate, like the one adopted in our work. However, such unconventional singularities are not present in the 3D ruptures with finite rupture width W that is the focus of our work. In the revised manuscript, we added new plots in Figure S6 and new text in Method A5 to highlight this difference between 2D and 3D rupture models:

This friction law produces a persistent and non-linear slip weakening similar to that caused by the lab-observed thermal weakening⁹², which obviously differs from the nearly-linear slip weakening produced by the regular rate-and-state friction with the aging law⁹¹. The resulting ruptures are pulse-like and we find that the slip rate in the tail of the pulse decays exponentially with distance (Figure S6e-f), which is fundamentally different than the power law decay that signals unconventional singularities in the 2D models by Brener and Bouchbinder⁹³, even with a similar friction law.

Moreover, in the present work we validated our 3D LEFM theory by showing that it explains well our simulation results with a complex weakening process similar to the enhanced dynamic weakening (Figure S6b): G_c agrees with G_0 within 3% in all the simulated steady-state models (Figure S2a). The validation of the 3D theory in this paper is original, innovative and promising. It can be extended in the future to other models with additional complexities, such as fluid effects and off-fault damage. To highlight the originality and essential features of our work, we modified the text as:

Numerical and theoretical studies^{53,54} demonstrated that **the energy balance governing rupture propagation on long faults with finite width in a 3D elastic medium depends on the rupture width rather than on the rupture length. In particular, Weng and Ampuero⁵³ developed a 3D theory of large ruptures that accounts for their finite rupture width W . Their theory yields a rupture tip equation of motion of the following form** (Methods A2):

$$F\left(\frac{G_c}{G_0}\right) = M(v_r) \dot{v}_r \quad (1)$$

where G_c/G_0 is the ratio between the fracture energy and the energy release rate of steady-state ruptures, \dot{v}_r is the rupture acceleration, the time derivative of rupture speed v_r , and F and M are known universal functions. **Here, the energy release rate of steady-state ruptures is exactly equal to the static energy release rate and, in contrast with the classical 2D fracture mechanics theory, it is independent of the rupture length⁵³: $G_0 = \Delta\tau^2 W/\pi\mu$, where $\Delta\tau$ is the static stress drop and μ is the shear modulus. These features of G_0 arise because the finite fault width W turns the rupture into a slip pulse with length smaller than W , and are similar to those of a rupture in a 2D strip of thickness W (ref^{55,56}). The finite W , an essential ingredient for the study of large earthquakes and SSEs, is missing in previous experimental and theoretical studies of slow and fast ruptures^{43,44}.**

...

Further linear stability analysis (Methods A2) shows that the general stability condition for steady-state ruptures can be written as

$$\frac{d(G_c/G_0)}{dv_r} > 0$$

The opposite condition, $d(G_c/G_0)/dv_r \leq 0$, predicts **non-steady ruptures** that either accelerate ($G_c/G_0 < 1$) or decelerate ($G_c/G_0 > 1$). **Generally, equation 1 predicts that there are two types of ruptures on long faults with finite width regardless of the rupture speed: steady and non-steady ruptures. The predicted features of non-steady ruptures with speeds typical of regular earthquakes (close to S-wave speed) have been numerically validated in a previous study⁵³, whereas the predicted steady-state ruptures of both fast and slow speeds are validated by numerical simulations here.**

...

The validations of equation 1 **for steady-state ruptures in this study and for non-steady ruptures in a previous study⁵³** prove that this equation can be used to describe both SSEs and earthquakes in a same theoretical framework, provided the dependence of the energy ratio, G_c/G_0 , on rupture speed, v_r , is known.

To acknowledge the discussion of the effects of fluid effects and off-fault damage, we added: **This fundamental theory can be extended in the future to other models with additional complexities, such as fluid effects and off-fault damage.**

References:

- Gabriel, A. A., J. P. Ampuero, L. Dalguer and P. M. Mai (2013). "Source properties of dynamic rupture pulses with off-fault plasticity." *Journal of Geophysical Research: Solid Earth* 118(8): 4117-4126.
- Paglialunga, F., F. Passelègue, N. Brantut, F. Barras, M. Lebihain and M. Violay (2021). "On the scale dependence in the dynamics of frictional rupture: constant fracture energy versus size-dependent breakdown work." arXiv preprint arXiv:2104.15103.
- Brantut, N. (2021). "Dilatancy Toughening of Shear Cracks and Implications for Slow

Rupture Propagation." Journal of Geophysical Research: Solid Earth 126(11).

Brener, E. A. and E. Bouchbinder (2021). "Unconventional singularities and energy balance in frictional rupture." Nature Communications 12(1).

2. Assuming that LEFM can generally be used to describe all types of frictional ruptures (slow-slip events to dynamic earthquake ruptures), what are the specific novel ideas presented in this work compared to prevailing views applying fracture mechanics for earthquake mechanics?

A. Prevailing views on fracture mechanics applied to earthquakes do not account for the finite rupture width W of large earthquakes and Slow Slip Events, which are longer than wide. An innovation in our work, is that we account for the finite rupture width W . This feature is missing in most previous theoretical and laboratory studies. As shown in Weng and Ampuero (2019), the effect of W on elongated ruptures changes drastically the scaling of G_0 and thus the rupture dynamics. Particularly, the energy release rate is independent of the rupture distance and rupture speed: $G_0 = \Delta\tau^2 W / \pi\mu$. That is fundamentally different from the classical theory, which deals with 1D or circular ruptures. Thus, fracture mechanics theories that do not include a finite W are insufficient to study large earthquakes and SSEs.

Another innovation in our work is that we have theoretically extended and numerically validated the 3D theory of Weng and Ampuero (2019) to friction that is not pure slip-weakening and to the full range of rupture speeds. We demonstrated that this new theory is suitable to describe the rupture propagation of both fast and slow earthquakes. To the best of our knowledge, no other team has yet proposed such extensions of this 3D theory, and no other theory has been proposed to describe 3D elongated earthquake ruptures. For further clarification of the originalities and innovations of our work, please refer to our response to question 1.

3. For readers who are not familiar with dynamic fracture theory, the current manuscript may seem quite technical. Following the arguments in the main text required continuous referral to the supplement (and the author's previous work, Weng and Ampuero JGR 2019) in order to follow along. For those who are more familiar with dynamic fracture theory and related literature, the key take-aways from the presented work seem to be a review of existing theoretical concepts and standard views of rupture growth. For example, it is well established for crack propagation in linear elastic theory that the balance between fracture energy and energy release rate controls rupture speed and crack growth, and an increase in fracture energy ahead of the crack tip is needed to balance the growing energy release in order to maintain a constant rupture speed (as described by many prior studies and discussed in detail in Freund, 1990 Dynamic Fracture Mechanics, Cambridge Monographs).

A. The "growing energy release" (with increasing rupture length) is actually a good example of a prediction of conventional 2D fracture mechanics theory, however, that is not valid in the presence of a finite rupture width W in what we called the 3D fracture mechanics theory.

Accounting for W is one of the originalities of our work. As a result of the finite W , the fracture energy and energy release rate of steady-state elongated ruptures are constant and are independent of the increasing rupture length (Figure S2a and Methods A5). Thus, we do not need to make this assumption, indicated by the reviewer, that “fracture energy increases with increasing rupture length to balance the growing energy release in order to maintain a constant rupture speed”.

We confidently argue that the key take-aways of this work are not a review of existing theoretical concepts and standard views of rupture growth. First, this paper generalized the two theoretical conditions for steady-state slow and fast ruptures on a long fault with finite width, extending the 3D fracture mechanics theory introduced by Weng and Ampuero (2019). Second, in this paper we numerically validated a number of theoretical predictions, which shows that the 3D LEFM can be used to describe both fast and slow large earthquakes. Please refer to our response to questions 1 by reviewer 1 for more details on the difference between the classical 2D theory and our new 3D theory. For further clarification of the originalities and innovations of our work, please refer to our response to question 1 by reviewer 2.

4. My understanding of the main arguments presented in the abstract, namely that a complete spectrum of rupture speeds is realizable for frictional rupture in nature and that the full range of earthquake and slow-slip transient behavior are fully described by traditional fracture mechanics, do not clearly appear to be supported the presented work.

A. Our work is not based on “traditional” fracture mechanics, but on our recent extension of fracture mechanics to ruptures with finite width W (see our reply to the two previous comments), which we referred to as “three-dimensional fracture mechanics” in the abstract. To clarify this, we modified the abstract as:

Here, we present numerical simulations that show that the rupture propagation of **both** slow slip events and earthquakes on long faults **with finite width** can be predicted by the **newly-developed** three-dimensional theory of dynamic fracture mechanics. **This model accounts for the finite rupture width, an essential ingredient of large slow and fast ruptures on natural faults, which has been missing in previous two-dimensional theories and experiments.**

The main conclusions presented in the abstract are supported in the paper in three different ways: by the results of our simulations, by our theoretical derivations and by our comparisons to real earthquakes and SSE observations.

5. However, there are some interesting points that could be potentially explored/discussed in detail, such as why existing observations don't appear to strongly suggest a full continuum of realized rupture speeds, but rather two groups of slow-slip or dynamic events.

A. We agree the rupture speed gap is an interesting feature of current observations. That is why we devoted a whole paragraph to this topic in lines 222-242 of the original manuscript, in which we introduced the observations and proposed two possible explanations based on the insight from our models.

6. and if the inferred global linear moment-duration scaling of slow slip events suggests a systematic variation in stress drops among regional populations of slow slip events.

A. We agree it is interesting to discuss the implication of different moment-duration scaling between slow slip events and earthquakes. Accordingly, we devoted two whole paragraphs to this topic in lines 180-207 of the original manuscript, in which we discuss the possible mechanism that may produce different moment-duration scalings.

7. How appropriate is the mathematical analogy between linear elastic dynamic fracture mechanics hold for frictional ruptures, particularly in terms of determining rupture speed?

A. The agreement between our 3D LEFM theory and our simulations (Figure 2b), including “long tailed” enhanced weakening, show that the LEFM approach is appropriate for frictional ruptures. Please refer to our response to question 1.

8. The presented work relies heavily on the assumption that standard linear elastic dynamic fracture mechanics applies to the broad array of frictional ruptures from slow-slip events to earthquake ruptures. While dynamic fracture theory works well for slip-weakening friction laws that are mathematically identical to cohesive zone laws of LEFM, it isn't clear how valid this analogy is for more realistic fault constitutive laws that include continuous weakening or variable dissipation behind the rupture tip. In other words, the work seems to limit itself to frictional formulations that are comparable to slip-weakening friction, though there is substantial evidence that such formulations may not be complete descriptions of fault friction. It is thus not clear whether the presented modeling results are specific to the chosen friction law and are necessarily relevant to natural slow-slip events and earthquakes.

A. The applicability of the classical 2D LEFM has been proven by previous works. It works well for slip-weakening friction laws but also for other weakening processes, such as enhanced dynamic weakening (e.g., Paglialunga, et al., 2021), off-fault plasticity (e.g., Gabriel, et al., 2013), fluid dilatancy and thermal pressurization (e.g., Brantut, 2020). Please refer to our response to question 1.

But our work does not rely on the assumption that the classical 2D LEFM is valid. On the contrary, this work presented a new 3D fracture mechanics theory that accounts for the finite rupture width and further demonstrated that this 3D theory is valid for describing the rupture propagation of both fast and slow earthquakes. In other words, the current work is new and complementary to the classical 2D theory.

In addition, the friction law assumed in our work produces a “continuous weakening” similar to that caused by thermal weakening. This can be appreciated by comparing our Figure S6b to Figure 3a of Paglialunga et al. (2021). Therefore, our work did not “limit itself to frictional formulations that are comparable to slip-weakening friction”, as the reviewer portrays. Our extended 3D fracture mechanics theory succeeds at explaining the numerical simulation

results, which demonstrates that our 3D LEFM with finite rupture width is not limited to slip-weakening friction. To clarify this difference, we revised the text in Methods A5:

The numerical simulations show that fault strength governed by V-shape rate-and-state friction has two weakening stages: the first stage accounts for the fast weakening process and the second stage accounts for the slow weakening process (Figure S6b). **This friction law produces a persistent and non-linear slip weakening similar to that caused by the lab-observed thermal weakening⁹¹, which obviously differs from the nearly-linear slip weakening produced by the regular rate-and-state friction with the aging law⁹⁰.**

...

We find that equation 29 predicts the relations among V_p , $\Delta\tau$, G_0 , and G_c well in all the simulated steady ruptures (Figure 2c & S2), **which demonstrates that the two weakening stages shall be considered as a whole cohesive zone in calculating the fracture energy.**

9. The mathematical analogy between singular cohesive-zone models of shear cracks and frictional ruptures depends on two key assumptions (Palmer and Rice, 1973):

(i) The breakdown of shear resistance during rupture is concentrated in a small region near the rupture front (i.e., small-scale yielding); and,

The (ii) A constant residual stress level exists throughout the ruptured region during sliding, such that the sliding stress level is comparable with the final stress level after rupture (allowing for this residual stress to be subtracted to draw analogy to cohesive zone models of mode I cracks).

In particular, the clear relationship between rupture speed and the fracture energy of linear elastic fracture mechanics is only valid under these assumptions.

A. In a previous paper (Weng and Ampuero, 2019), the 3D fracture mechanics theory that accounts for the finite width W was numerically validated for non-steady ruptures controlled by linear slip-weakening friction. In our following works, we extended this validation to an exponential slip-weakening friction law (these results will be submitted for publication in a specialized journal this year). In these works, the relationships between the rupture speed and the fracture energy is consistent with the 3D theoretical prediction, because the two key assumptions mentioned by the reviewer are satisfied: the cohesive zones are much shorter than the minimum rupture dimension W and the dynamic frictional strength is comparable with the final stress.

In this paper, the 3D fracture mechanics theory is extended and numerically validated for steady-state ruptures controlled by a complex weakening/strengthening behavior (Figure S6), which also satisfies these two key assumptions. The direct evidence is that the lengths of the pulse-like ruptures, caused in our models by the effects of the finite rupture width, are much shorter than the minimum rupture dimension W (Figure S6c-d), as also demonstrated in previous numerical simulations (Figures 5d in Hawthorne and Rubin, 2013). In addition, in our current numerical simulations, we validated that the relationship between the rupture speed and the fracture energy is consistent with the 3D theoretical prediction (Figures 2).

Reference:

Hawthorne, J. and A. Rubin (2013). "Laterally propagating slow slip events in a rate and state friction model with a velocity-weakening to velocity-strengthening transition." *Journal of Geophysical Research: Solid Earth* 118(7): 3785-3808.

10. This work explores rate-and-state friction including some rate-strengthening at high slip rates, "as observed in laboratory experiments" (lines 49-51), which is generally consistent with slip-weakening friction.

A. Not quite. As shown in Figure S6 of the original manuscript, the assumed friction law produces two weakening stages **similar to that caused by the lab-observed thermal weakening**. It is not a trivial slip-weakening.

11. The work does mention that enhanced dynamic weakening can occur at high slip rates, which in fact is widely documented in laboratory experiments (e.g. Di Toro et al., Nature 2011), however there is no discussion of how this alters the underlying assumption that traditional dynamic fracture theory is valid.

A. We do not assume the applicability of the traditional 2D theory, which has been validated by other studies. Instead, we adopted and extended a newly-developed 3D fracture mechanics theory that accounts for the finite rupture width W and validated it by numerical simulations. Please refer to our response to question 1.

In addition, enhanced dynamic weakening produces an approximately exponential slip-weakening curve with a "long-tailed" weakening (e.g. Di Toro et al., 2011), which is qualitatively similar to the curve presented in this work (Figure S6b). Because we have numerically validated the small-scale yielding approximation for the "long-tailed" weakening, we believe that this theory shall also be valid for thermal weakening at seismic rates. Our optimism comes from our work in progress, which has already extended the friction to the exponential slip-weakening law (these results will be submitted for publication in a specialized journal this year). To clarify this, we added the following discussion:

At larger slips, **previous studies suggested⁶⁰⁻⁶³ that** weakening controlled by thermal pressurization or by off-fault inelasticity (bulk plasticity or damage) may be slip-dependent rather than rate-dependent, **which remains to be confirmed by future studies accounting for the finite rupture width. Assuming these slip-dependent features, G_c/G_0 is a function⁴² of the final slip (D) independent of v_r , $G_c/G_0 \sim D^{n-2}$, where $n = 2/3$ for thermal pressurization⁶¹ and $n = 1$ for theoretical off-fault inelastic dissipation⁶³.**

12. Recent numerical studies have demonstrated that both of these key assumptions become invalid with the inclusion of thermo-hydro-mechanical processes, such as the thermal pressurization of pore fluids, which has been suggested to be prevalent for nature earthquakes of varying size (Abercrombie and Rice, JGR 2006; Viesca and Garagash, Nature Geo 2015; Lambert and Lapusta, Solid Earth 2020). Enhanced dynamic weakening can result in continued or variable weakening well behind the rupture front and the associated local

breakdown energy (thought to be the frictional equivalent to fracture energy) is rupture-dependent, i.e. not a fixed property of the interface. In such case, the original analogy to traditional fracture theory is no longer obviously justified and the relationship between breakdown energy and rupture speed is not clear.

A. Please refer to our response to question 1.

13. Moreover, the relationship between the energy balance of frictional ruptures and the standard energy budget based on dynamic fracture theory depends on the rupture style, for example self-healing pulse-like ruptures are not well-described by the standard energy budget (Heaton, PEPI 1990; Viesca and Garagash, Nature Geo 2015; Lambert et al., Nature 2021).

A. This criticism is demonstrably invalid: all the ruptures in our current simulations and in those we published previously (Weng and Ampuero, 2019, 2020) are pulses, owing to the effect of W , and yet our extended fracture mechanics theory has succeeded in explaining those simulation results.

14. Note that other mechanisms for variable breakdown energy during dynamic rupture, such as production of damage and off-fault inelastic deformation (e.g. Andrews, JGR 2005; Okubo et al., JGR 2019) would likely also violate these assumptions as the relevant dissipation for the dynamic is not constrained to the rupture tip.

A. Fracture mechanics has been successfully used to interpret dynamic rupture simulations with off-fault dissipation, notably in Figure 10 of Andrews (2005) and in Gabriel et al (JGR 2013).

15. Overall, how does the presented work account for or comment on the accumulating body of work that question the underlying assumptions for the analogy between traditional dynamic fracture theory and frictional ruptures?

A. As shown in multiple replies above, our extended 3D fracture mechanics theory works well for slow and fast large ruptures with finite width.

16. On a related note, does the presented modeling reproduce the well-known inferred increase in average breakdown energy with average slip and/or rupture size of earthquakes, as determined for natural earthquakes and in laboratory experiments (Abercrombie and Rice, JGR 2005; Rice, JGR 2006; Cocco and Tinti, 2008; Viesca and Garagash, 2015; Nielsen et al., 2016; Brantut and Viesca, 2018; Selvadurai, JGR 2019). Such inferences have typically been used to suggest that thermal pressurization or some other dissipation (such as along off-fault damage) are prevalent across earthquakes of a broad range of sizes, unless some special heterogeneity exists along the fault.

A. Those observations are accounted for in our models via the scaling of G_c with slip.

17. Assuming that linear elastic dynamic fracture theory can be appropriately applied to frictional ruptures in general, what novel insight is supported by the presented modeling?

The balance between fracture energy and energy release rate in determining rupture speed is a well-established aspect of dynamic fracture theory and fracture mechanics has been widely applied to study aspects of earthquake ruptures (e.g. Palmer & Rice, 1973; Madariaga, 1976; Kostrov and Das, 1998; Kanamori and Brodsky, 2004; Rubin and Ampuero, 2005). The proposal that fracture mechanics works well for describing ruptures governed by rate-and-state friction is not really novel, and the generality of its strict application is questionable as noted above.

A. The key originalities of our work are summarized in our reply to comment 2. The applicability of our extended 3D LEFM is documented in multiple replies above.

18. Key point #1 from the abstract (lines 13-15) is that a “complete spectrum of rupture speeds Is quantitatively controlled by the ratio of fracture energy to energy release rate, which depends on the frictional behavior on the fault.”

This is a well-established aspect of dynamic fracture theory but would not clearly be evident when small-scale yielding is violated, as mentioned earlier. Moreover, the energy release rate may not just be a function of frictional behavior along a fault, but weakening or strengthening can depend on other mechanisms such as damage generation or enhanced dynamic weakening such as from thermal pressurization. In the case of enhanced weakening, large dynamic stress changes due to efficient thermal pressurization would depend on the dynamics of the individual rupture and increase the energy release rate and may not be strictly related to the static stress changes: (Heaton, PEPI 1990; Lambert et al. Nature 2021).

A. Please refer to the above responses.

19. Key point #2 from the abstract (line 16-18) addresses the difference in linear versus cubic moment-duration scaling of slow-slip events, where heterogeneity in shear stress produces cubic scaling while heterogeneity in effective normal stress produces linear scaling.

The presented arguments for the cubic scaling for ruptures with heterogeneous shear stress but constant rupture speed were not particularly clear in this work. In essence, it would seem that this argument really follows the results of Dal Zilio et al., GRL 2020, who demonstrated that rupture speed can vary substantially during slow-slip events due to heterogeneous shear stress conditions in finite ruptures and accounting for this variable rupture speed reproduces the cubic moment-durations scaling. It is not clear how a similar result arises from scaling arguments with a constant rupture speed.

A. We do not assume constant rupture speeds in our arguments. For the models with heterogeneous shear stress, the rupture speed scales with the rupture length (Figure S3b).

20. 1. Spectrum of rupture speeds:

The proposed results from dynamic fracture theory predict a spectrum of rupture speeds is possible depending on critical strengthening velocity. However, a complete spectrum of

rupture speeds may not actually be observed, and hence realized.

A. A complete spectrum of rupture speeds has been realized and observed in multiple laboratory experiments, such as Passelegue et al. (2020), Im et al. (2020), Svetlizky et al. (2017), Kammer et al. (2018), etc. We nevertheless acknowledged and discussed the possibility of a speed gap in natural faults; please see our reply to comment 5.

References:

Passelègue, F. X., M. Almakari, P. Dublanchet, F. Barras, J. Fortin and M. Violay (2020). "Initial effective stress controls the nature of earthquakes." *Nature communications* 11(1): 1-8.

Im, K., D. Saffer, C. Marone and J.-P. Avouac (2020). "Slip-rate-dependent friction as a universal mechanism for slow slip events." *Nature Geoscience* 13(10): 705-710.

Svetlizky, I., D. S. Kammer, E. Bayart, G. Cohen and J. Fineberg (2017). "Brittle Fracture Theory Predicts the Equation of Motion of Frictional Rupture Fronts." *Physical Review Letters* 118(12): 125501.

Kammer, D. S., I. Svetlizky, G. Cohen and J. Fineberg (2018). "The equation of motion for supershear frictional rupture fronts." *Science advances* 4(7).

21. Figure 2a demonstrates a potential continuum of rupture speeds in the modeling, however existing inferences from natural events in Figure 1b don't actually seem to demonstrate a continuum. How evident is it that there is a "continuum of rupture speeds" for slow-slip to dynamic earthquake ruptures? It would appear that all of the earthquake rupture speeds are concentrated in one group (dynamic events) and then slow-slip events are clustered together as well, with the large gap of several orders of magnitude in slip rate and rupture speed being somewhat filled with a few inferences from Izu-Bonin SSEs and some lab experiments. Is it possible that there is not actually a continuum of plausible rupture speeds but instead two favorable regimes?

A. This point is addressed in our reply to comment 5.

22. Two potential avenues are suggested by this work: 1) More observations are needed to fill in this gap; 2) the critical velocity is low enough that regions are separated into two different stability regimes

Can the author comment more on the relationship between this critical velocity and different physical mechanisms (e.g. strength of dilatancy vs thermal pressurization) to suggest why there may not be a realized continuum of rupture speeds but ruptures are clustered as either slow-slip or dynamic events? On lines 217-218 the author claims that this critical slip rate can be "constrained" with theoretical arguments, however this range is essentially the full range of slip rates from around typical tectonic plate rates to near seismic slip values. Can a more informative constraint be made with these considerations that a continuum may not be realized?

A. We devoted a paragraph to discuss the relationship between this critical velocity and different physical mechanisms (lines 83-112 of the original manuscript) and another

paragraph to discuss the gap of the rupture speed (lines 222-242 of the original manuscript). The continuum range of critical slip rates is observed in laboratory experiments cited in Figure 2d. A maximum critical slip rate in natural faults is one possible explanation constrained by the observed gap of rupture speed.

23. The friction law used includes some rate-strengthening above a critical value, presumably consistent with models of gouge dilatancy during the initial acceleration of slip?

A. Gouge dilatancy is another possible mechanism that can produce the rate strengthening that is discussed in lines 86-88 of the original manuscript.

24. Do any laboratory experiments show continued strong rate-strengthening at seismic slip rates?

A. To our knowledge, there is no evidence of strong rate-strengthening at seismic slip rates. The reason we considered the continued strong rate-strengthening at seismic slip rates in Figure 2 is to validate the steady-state version of equation (1). We have devoted a paragraph and Figure 1c to discuss the rupture behaviors controlled by specific frictional laws. For example:

At larger slips, **previous studies suggested⁶⁰⁻⁶³ that** weakening controlled by thermal pressurization or by off-fault inelasticity (bulk plasticity or damage) may be slip-dependent rather than rate-dependent, **which remains to be confirmed by future studies accounting for the finite rupture width. Assuming these slip-dependent features, G_c/G_0 is a function⁴² of the final slip (D) independent of v_r , $G_c/G_0 \sim D^{n-2}$, where $n = 2/3$ for thermal pressurization⁶¹ and $n = 1$ for theoretical off-fault inelastic dissipation⁶³.**

25. How would your results differ if you included a second critical velocity with a transition in weakening behavior, such as strong rate weakening friction of flash heating or thermal pressurization (Rice, JGR 2006)?

A. This is discussed in lines 83-112 and Figure 1c of the original manuscript.

26. 2. Moment-duration scaling of slow slip events

The main argument seems to be that previous work (e.g. Dal Zilio et al., GRL 2020) has demonstrated that cubic scaling is possible when accounting for spatially variable stress changes and rupture speeds during slow-slip events, however if one combines multiple populations with cubic scaling then the relation can appear linear if stress drop varies among populations.

Comparing regional slow-slip population moment-duration scaling in addition to global scaling seems to be a potentially insightful practice. It would seem that the argument here for an overall linear global trend may suggest a systematic difference in stress drop among slow-slip event populations. Do observations from natural slow-slip events support this, i.e. do some slow-slip event populations have systematically lower or higher average stress drops

consistent with this shift to make a linear scaling?

A. Yes, there is a systematic difference in the estimated stress drop among different SSE populations. We added a new Figures S3c and S3d and new discussions in the main text:

However, if the shear stress distribution on the fault is heterogeneous, and in particular if it decays linearly away from the nucleation area (Methods A8), the simulated models result in $\alpha = 0.5$ and $\beta = 0.5$, which leads to a cubic scaling relation (Figures 4a & S3), **consistent with the observations in the Cascadia subduction zone (Figures S3c & S3d).**

...

This can explain the observed linear scaling based on a global compilation of various slow earthquakes from different fault environments whose σ may span several orders of magnitude, such as $\sigma \sim 1 - 10$ kPa on the deep San Andreas fault⁷⁸ and $\sigma > 1$ MPa in the Izu-Bonin Trench¹⁶. **The explanation based on the diverse values of σ is also supported by the diverse values of stress drop of the observed SSEs, which span at least 3 orders of magnitude (Figure S3c).**

Figure S3: **Scaling relations of stress drop and rupture speed.** Stress drop (a) and rupture speed (b) as a function of rupture length for the homogeneous model (pink circles) and the linear decay model (green triangles). (c-d) Stress drop and rupture speed of real SSEs from three different regions (indicated in legend) versus normalized rupture length. The stress drop is simply estimated by $\Delta\tau \approx \mu D/W$.

27. Note that the line numbers are missing from parts of the pdf which makes it challenging to specifically reference lines in the text.

A. Revised.

28. Figure 3b: It isn't clear what the x and y axes are in the inset.

A. Revised.

29. Lines 92-93: “weakening controlled by thermal pressurization ... is slip-dependent rather than rate-dependent.” This is not true. Recent numerical studies (Lambert and Lapusta, Solid Earth 2020) including thermal pressurization have demonstrated that weakening behavior due to thermal pressurization is not a unique function of slip but depends on the history of sliding rate and dynamic stress changes throughout individual ruptures. The same point along a fault can experience different weakening behavior with comparable amounts of slip in different ruptures due to differences in the slip rate function and dynamic stress changes. This in turn means that local breakdown energy is 1) rupture-dependent and 2) does not strictly increase or decrease with local slip. The pure slip-dependence of thermal pressurization in early theory assumes a constant sliding rate, or a consistent slip rate function at all points along a fault, which is not the case in finite ruptures.

A. To clarify this, we modified the text as:

At larger slips, **previous studies suggested⁶⁰⁻⁶³ that** weakening controlled by thermal pressurization or by off-fault inelasticity (bulk plasticity or damage) may be slip-dependent rather than rate-dependent, **which remains to be confirmed by future studies accounting for the finite rupture width. Assuming these slip-dependent features, G_c/G_0 is a function⁴² of the final slip (D) independent of v_r , $G_c/G_0 \sim D^{n-2}$, where $n = 2/3$ for thermal pressurization⁶¹ and $n = 1$ for theoretical off-fault inelastic dissipation⁶³.**

30. Line 96: Why is D necessarily bounded by the interseismic slip deficit for individual points during the rupture? It is well demonstrated in dynamic rupture models that points throughout ruptures can exhibit a dynamic overshoot and slip more than would be expected in a quasi-static calculation based on the accumulated static stress to drop. A point can overshoot slip and then account for this with extra locking during the subsequent interseismic period.

A. Previous dynamic rupture simulations, primarily based on slip-weakening friction, show

that overshoot is limited to about 20%. This estimate is not large, so for simplicity we ignored it in this discussion.

31. Lines 167-171. The inferred low-stress operation of mature faults is with respect to typical Byerlee values of friction (~ 0.6) consistent with steady-state friction coefficients measured in the lab (perhaps consistent with the shear stress conditions for nucleation), not the peak friction, which could in fact be much higher. Note that the peak shear stress depends on how well-healed the fault is through the state variable as well as the slip rate during rupture. Given the mild logarithmic rate-weakening of standard rate-and-state friction (as used in this work), the average stress state along the rupture area is within one static stress drop of this quasi-static strength for nucleation, so such models (including those presented in this work) would need either low quasi-static reference friction or the fault would need to have low effective normal stress, such as through substantial fluid overpressure, to be compatible with the inferred low effective friction conditions.

A. The whole paragraph that contains Lines 167-171 is based on the assumption of the thermal weakening observed in laboratory experiments, rather than the logarithmic rate-weakening of standard rate-and-state friction. For thermal weakening, the critical slip-weakening distance (or the thermal slip distance as defined in Di Toro et al., 2011) is short, therefore the average apparent fault strength can be much lower than the static fault strength. That is the “statically strong, dynamically weak” paradigm of Jim Rice and co-workers. Combining thermal weakening and the 3D rupture mechanics theory with finite rupture width W , we found that the apparent fault strength (the average shear stress at which the fault operates) decreases with increasing effective normal stress. To clarify that, we added: Note that the apparent fault strength is derived based on the assumption of the thermal weakening with exponential slip-weakening as observed in laboratory experiments⁶² and the complexity of fault roughness⁷⁶ is not considered.

32. Equation (16) the meaning of L_c is not defined – process zone size

A. Revised.

33. Rupture time is computed for the numerical models by the slip rate criterion $V = 10V_i$, so this depends on the choice of V_i ? Have you investigated the sensitivity of timing to choice of threshold? Particularly for slow-slip events where slip rates and slip acceleration can vary substantially?

A: To calculate the rupture speed, it is necessary to define a threshold of slip rate for the rupture tip. We have verified that the choice of V_i does not affect the results much.

34. Doesn't Equation (17) assume a uniformly prestressed crack or constant stress drop? Do you compute energy release rate directly in your simulations, or use this equation?

A. Equation (17) assumes a uniform stress drop for steady-state ruptures. We compute the

energy release rate of all steady-state rupture simulations by using this equation as explained in Methods A5. For heterogeneous stress drop, please see equation 55 in Weng and Ampuero (2019).

35. Here, the energy release rate is considered a function of static stress drop once again in the analogy to cracks where a traction-free surface occurs behind the crack tip and essentially all break down or dissipation occurs at the rupture tip. In other words, the static stress drop is a reasonable measure of the total stress change during the dynamic process. For frictional ruptures, dissipation can occur throughout the entire rupture and the energy release rate would be related to the difference between the total strain energy change and the energy that is dissipated, along the fault or in creating new fractures off-fault such as with damage. So the energy release rate is not strictly related to the stress drop but would be a more complicated consideration of the dynamic stress changes. For example, the static stress drop could vastly underestimate the dynamic stress changes in the case of a self-healing pulse with a substantial stress undershoot (e.g. Lambert et al., Nature 2021).

A. The energy release rate formula in equation (17) is mathematically exact for the specific problem stated in the manuscript, steady-state rupture propagation with finite rupture width W and uniform stress drop, as rigorously derived by Weng and Ampuero (2019). For heterogeneous stress drop $\Delta\tau(x)$, the general solution involves a weighted integral of stress drop (equation 55 in Weng and Ampuero, 2019). As the weighting function decays sharply with distance behind the rupture tip, the energy release rate is mainly controlled by the values of $\Delta\tau(x)$ shortly behind the rupture tip. The reviewer's view that the energy release rate shall depend on the whole history of the ruptured area is correct in 2D crack ruptures, but not in 3D pulse-like ruptures on a long fault with finite width.

36. Equation (18) assumes that the final level of shear stress is equal to the minimum shear resistance during sliding, which would not be the case for an overshoot or undershoot, for example in the case of a self-healing pulse (Heaton, PEPI 1990, Viesca and Garagash, Nature Geosci 2016, Lambert et al., Nature 2021).

A. Equation (18) does not make such an assumption. Actually, equation (18) is the general definition of the fracture energy (see equation 5.3.19 in Freund, 1990), which accounts for the whole frictional history: an integral from the start level to the final level of shear stress (see Figure S6b).

For rate-and-state friction with mild slip rate dependence of the shear resistance, the level of sliding resistance and final stress may be comparable, with mild overshoot. However this need not be the case for enhanced dynamic weakening, as may be expected to dominate during earthquake ruptures.

A. In our case LFM works when we account for the second-weakening as a contribution to G_c . This works because, as explained in a previous reply and new fig S6e-f, in the tail of our pulses the slip rate decays exponentially, steeper than any power-law. In particular, the decay

in our 3D problem with W-effect is steeper than the unconventional singularities that have recently put forward to question the validity of LEFM in 2D problems with a similar friction law but without W-effect.

37. Before equation (26), $T_r - T_f$ is not the traditional definition of overshoot, which is rather the stress adjustment due to waves after slip has arrested. This quantity $T_r - T_f$ would just be continued weakening during slip within the rupture surface. From dynamic fracture theory It isn't clear how relevant this would be for driving the rupture tip.

A. We have revised it as: τ_f is the final fault strength after rupture arrest and τ_r is the fault strength at the tail of the first weakening stage.

38. How is stress drop being estimated in these models? Is it uniform? The spatial average?

A. For steady ruptures, the stress drop is uniform along the fault. For the heterogeneous models for the M-T scaling, the stress drops are the spatial averages.

39. Is this energy release rate assuming uniform stress drop throughout the rupture? This would certainly not be the case natural ruptures, which is even evidenced from finite-fault inversions (e.g. finite-fault inversions of large earthquakes in Ye et al., JGR 2016ab). Does this critical stress drop for steady propagation also assume that stress drop is uniform within the rupture area?

A. For the steady-state ruptures, we assume uniform stress drops. But for a general solution of the energy release rate, please see equation 55 in Weng and Ampuero (2019). The critical stress drop depends on the friction parameters of the fault. Please see Appendix A7 for the analytical derivations.

40. Line 364: In the homogeneous model where the stress drop > critical value, ruptures steadily propagate throughout the entire fault. Does this mean that the ruptures don't naturally arrest but are forcibly arrested by the model geometry, i.e. the VS boundaries?

A. Yes, these ruptures are stopped by a strong barrier, which can be a VS boundary or a VW boundary with very low shear stress.

How do the average rupture properties – duration, stress drop, rupture speed – depend on the properties of the VS boundaries?

A. Line 364 discusses a special case of steady-state ruptures that can only be stopped by a strong barrier. In this special case, rupture speed and stress drop are scale-independent while rupture duration and moment scale with rupture length (by varying the location of the barrier). Because this special case produces a similar linear scaling as cases with stress drop < critical value (Figure 4a), we only discuss it in Appendix A8.

If one wants to compare scaling relations among aspects of the full finite rupture then accounting for these regions of rupture propagation and arrest would seem to be important, particularly if the VS boundaries are not very strong and propagation continues substantially into these regions (?).

A. Accounting for a mild barrier will increase the complexities of the current model, whereas here we aimed to reveal the first-order factor that may control the M-T scaling relation. Actually, the quantitative investigation of the kind of model the reviewer suggests has been conducted and presented in Figure 3b.

41. Equation (31): Presumably the factor of $(1-\nu)$ in the “strike-slip” term should be related to mode II rupture (vs μ for mode III), which for a megathrust fault would be dip-slip.

A. In this paper, we focus on the dip-slip ruptures, which represent most of the observed SSEs.

Comments from reviewer #3:

1. The word 'steady-rupture' appeared lots of times but I don't understand. Earthquakes start from acceleration in slip and then decrease to die out. The process is sensitive to not just structural heterogeneities but also various physics such as thermal weakening and pore fluid changes as also mentioned in the manuscript. I don't understand how does 'steady' explains the process. Maybe the author means 'sustain-rupture'? Either change the word or explaining it properly in the context.

A. A steady-state rupture model is a canonical model to study rupture dynamics, i.e. a simplified model with strengths and weaknesses: despite being incomplete, it captures important aspects of the process, provides valuable insights and is a useful reference for more complex modeling. Indeed, in this work we take this simple model as a starting point, and later in the paper we introduce complexities such as heterogeneous stress. To further clarify the steady and non-steady ruptures, we added these in the main text:

Generally, equation 1 predicts that there are two types of ruptures on long faults with finite width regardless of the rupture speed: steady and non-steady ruptures. The predicted features of non-steady ruptures with speeds typical of regular earthquakes (close to S-wave speed) have been numerically validated in a previous study⁵³, whereas the predicted steady-state ruptures of both fast and slow speeds are validated by numerical simulations here.

2. Some quantitate in the research are not defined clearly. For example, the definition of stress drop could be various depending on the averaging methods (refer to 3.1-3.3 in Perry et al. 2021).

A. The static stress drop is uniquely defined as the difference of shear stresses before and after the ruptures.

There is quite a variation in the 2D forward model and larger variations should be expected in 3D model.

A. For steady-state ruptures, all quantities are uniform along the fault. For the heterogeneous ruptures presented for the M-T scaling, the average stress drop is calculated by averaging the stress drop on the ruptured region.

Critical slip rate V_c is an important parameter in the analysis, however, it is not clear how V_c is determined in the model and how it is supported by experiments.

A. To clarify the critical slip rate, we modified the text as:

Here, we further validate equation (1) for a continuum of **steady** rupture speeds from arbitrarily-slow speeds up to the S-wave speed **in numerical simulations** (Figure S1A). **This model considers** a rate-and-state friction law with rate-weakening behaviour at low slip rates

and rate-strengthening behaviour **above a critical slip rate** (Methods A3), as observed in laboratory experiments³¹⁻⁴¹. **We find that the finite-width fault, controlled by a rate-dependent friction, also turns the dynamic rupture into a slip pulse (Figure S6c&d), which is consistent with previous quasi-dynamic simulations²³. The critical slip rate is an important quantity directly measured in the laboratory experiments (Figure 2d) and such slip-rate-dependent friction behavior has been suggested to be a mechanism for slow slip events³¹.**

Why don't use the more standard RSF format? Will it result in different tuning parameters?

A. The slip-rate-dependent friction behavior has been widely reported in laboratory experiments (citations in Figure 2d) and has been suggested to be a universal mechanism for slow slip events (e.g., Im et al., 2020). The “standard format” of RSF, an empirical equation that assumes either rate weakening or rate strengthening at all slip rates, cannot account for the observed friction transition from rate weakening to rate strengthening.

Reference:

Im, K., D. Saffer, C. Marone and J.-P. Avouac (2020). "Slip-rate-dependent friction as a universal mechanism for slow slip events." *Nature Geoscience* 13(10): 705-710.

3. The 2.5D model uses a uniform velocity model (at least I guess so since it is not mentioned of the velocity model in A4), which will ignore the effects of material rigidity. It has been shown in a lot of studies that 3D structure will affect the rupture speed, especially for shallow tsunami earthquakes (e.g. Lay et al. 2012). The source depths of SSEs also span a wide range of temperatures and materials which play an important role. There is, however, only one sentence on line 251 discussing this effect.

A. The models in this paper use a uniform velocity model; we did not consider the effect of the 3D velocity structure explicitly. Nevertheless, the effect of shear modulus μ is included in our theoretical derivations (Appendix A5) and is encapsulated in the nondimensionalized critical slip rate $V_c \mu / \sigma v_s$.

4. The discussion of the moment-duration scaling part is interesting. The author attributes the “debated scaling behaviors can be attributed to different types of fault heterogeneities: heterogeneity of shear stress can produce a cubic scaling, whereas heterogeneity of effective normal stress produces a linear scaling.” What implication on tectonic stress environments can be learned from these heterogeneities? What kind of volumetric tectonic loading generates only shear stress or normal stress heterogeneity?

A. The heterogeneity of shear stress has been widely reported in cycle simulations even when prescribed friction properties are uniform (e.g., Dal Zilio et al., 2020). The variation of effective normal stress on different natural faults for slow earthquakes has also been reported. For example, the constrained effective normal stress is 1-10 kPa on the deep San Andreas fault⁷³ and $\sigma > 1$ MPa in the Izu-Bonin Trench¹⁶, which is attributed to differences of fluid

pressure. The implications have been discussed in the original manuscript. Other heterogeneities, such as the material rigidity and the fault width, may also affect the scaling relation. But we found that in the theoretical model, only the effective normal stress produces a linear envelope trend when varied, because M_0 and T have the same scaling dependence on that parameter but not on others.

Reference:

Dal Zilio, L., N. Lapusta and J. P. Avouac (2020). "Unraveling Scaling Properties of Slow-Slip Events." *Geophysical Research Letters* 47(10): e2020GL087477.

5. The introduction is not quite related to the framework of the paper. I would suggest reshaping it a bit to explain what is the correct research gap between slow and fast earthquakes and why is the linear or cubic scaling between seismic and aseismic events important. Which research questions does this work address?

A. An innovation in our work, compared to the previous body of work applying 2D fracture mechanics to earthquakes, is that we account for the finite rupture width W . This feature is missing in most previous theoretical and laboratory studies. As shown in Weng and Ampuero (2019), the effect of W on elongated ruptures changes drastically the scaling of energy release rate and thus the rupture dynamics. Thus, fracture mechanics theories that do not include a finite W are insufficient to study large earthquakes and SSEs. Another innovation in our work is that we have numerically validated this 3D theory and demonstrated that it is available to describe the rupture propagation of both fast and slow earthquakes even when the frictional behavior of the fault is complex. The reason the M-T scaling is important is that such empirical scaling has been used to constrain the physical mechanism of slow slip events, as mentioned in the original manuscript.

To highlight the innovation of this paper, we modified the abstract by

Here, we present numerical simulations that show that the rupture propagation of **both** slow slip events and earthquakes on long faults **with finite width** can be predicted by the **newly-developed** three-dimensional theory of dynamic fracture mechanics. **This model accounts for the finite rupture width, an essential ingredient of large slow and fast ruptures on natural faults, which has been missing in previous two-dimensional theories and experiments.**

and we modified the introduction by

Though laboratory experiments^{22,34,43,44} have suggested a continuum of rupture speeds, **these experiments lack a finite rupture width, an essential ingredient of large slow and fast ruptures on natural faults**, and the general rupture mechanics **controlled by such finite rupture width** is not completely understood.

6. line 93: the expression is not correct.

A. We revised these sentences as:

At larger slips, **previous studies suggested⁶⁰⁻⁶³ that** weakening controlled by thermal pressurization or by off-fault inelasticity (bulk plasticity or damage) may be slip-dependent

rather than rate-dependent, **which remains to be confirmed by future studies if accounting for the finite rupture width. Assuming these slip-dependent features, G_c/G_0 is a function⁴² of the final slip (D) independent of v_r , $G_c/G_0 \sim D^{n-2}$, where $n = 2/3$ for thermal pressurization⁶¹ and $n = 1$ for theoretical off-fault inelastic dissipation⁶³.**

7. line 96: I don't think it is a valid assumption, considering only partial ruptures and the less well-constrained slip deficit inversion. A recent example could be 2011 Tohoku-oki event where shallow less coupled fault ruptured with maximum slip.

A. Conventional inversions of subduction seismic coupling based on inland GPS data are poorly constrained at shallow depth, far from the coast. A recent analysis introducing physics-based constraints in the inversion demonstrated that the shallow Tohoku megathrust has a high coupling: a slip rate deficit between 80 and 100% of the plate convergence rate (Lindsey et al., 2021). Therefore the maximum slip of the 2011 Tohoku-oki earthquake might as well have occurred in a highly coupled shallow region. We thus believe our assumption that “D [the slip of characteristic earthquakes] is bounded by the interseismic slip deficit” is reasonable, as long as a properly resolved coupling map is available. To clarify it, we revised this sentence as

Therefore, considering that D is bounded by the interseismic slip deficit, **given a coupling map with sufficient resolution⁶³**, the larger the accumulated slip deficit on the fault, the smaller G_c/G_0 can be **(as a worst-case scenario)**.

Reference:

Lindsey, E. O., R. Mallick, J. A. Hubbard, K. E. Bradley, R. V. Almeida, J. D. P. Moore, R. Bürgmann and E. M. Hill (2021). "Slip rate deficit and earthquake potential on shallow megathrusts." *Nature Geoscience* 14(5): 321-326.

8. line 104: I don't think it is correct. Segall and Bradley shows that the repeated SSEs produce a stress concentration at the top of the ETS zone which causes thermal pressurization that is sufficient to allow rupture to accelerate to inertial limits. I find, however, this is not consistent with the statement that “this steady SSE could transition to a non-steady earthquake and accelerate toward the S-wave speed”.

A. Line 104 is correct. Segall and Bradley showed that one mode by which SSEs can lead to an earthquake involves an SSE propagating updip and undergoing continuous rupture acceleration to seismic speeds when the accumulated slip deficit is sufficiently high near the top of the SSE zone (see Figures 2a and 3a in Segall and Bradley, 2012). Line 104 states that when the rupture propagates from a region with low slip deficit to a region with high slip deficit, the rupture can accelerate to seismic speeds, which is an accurate description of Segall and Bradley's result.

Additionally, the model here where G_0 is constant and not changeable during cycles?

A. G_0 changes during cycles.

9. line 108-109: I don't understand what is need to be confirmed.

A. Our newly-developed theory including a finite rupture width W predicts that there are “two admissible steady speeds if the two competing frictional mechanisms are comparable (dash red curve in Figure 1c)” (lines 107-108). One is a slow speed and the other is seismic, close to S-wave speed. We “propose that the rupture is more likely to accelerate toward the S-wave speed” (lines 108-109). To our knowledge, there is no numerical simulation that combines these two frictional mechanisms and a finite fault width, therefore the hypothesis proposed in lines 108-109 remains to be tested in future simulations.

10. line 141: effective normal stress should be $\bar{\sigma}$

A. σ is defined as the effective normal stress everywhere in this paper, so to keep it concise we prefer to use σ .

11. line 142: I don't think it is a stress drop, might be slip? The formula might be wrong since a parenthesis is missing for $\pi \mu$

A. It is defined as “critical final slip” in the same sentence (line 141). We added parentheses to disambiguate the equation.

12. lines 150-153: this part is not new and not well related to the paper

A. The theoretical estimates in this sentence are derived from our newly-developed 3D theory that accounts for the finite rupture width. They are intended to support that the new model discussed in this paragraph is consistent in order-of-magnitude with real observations.

13. Line 170: I don't think it is correct. The author compares $\mu_r + 0.5 \cdot \text{average stress drop} / \text{normal stress}$, which equals to $\tau_0 / \text{normal stress}$, and since τ_0 is always smaller than peak τ (τ_p) in RSF, the comparison doesn't have anything to do with static fault stress (τ_p is not τ_{static}).

A. τ_p is the peak frictional strength, which we consider to be comparable to Byerlee's friction (“static friction”). Our point is not simply that τ_0 is smaller than τ_p , that would be obvious; but that τ_0 is **much** smaller than τ_p (we used the symbol \ll).

Additionally, I don't think this derivation of the relation between fault average strength and static strength is novel in this study.

A. While the concept that dynamic weakening allows faults to operate at low stress is not new and has been abundantly demonstrated in published simulations, we are confident that the derivation proposed here is original because it is based on a new-developed 3D theory that

accounts for the finite rupture width (Weng and Ampuero, 2019). The other analytical derivation of average stress we are aware of is the work by Eric Dunham's team based on a rough fault model, which is a fundamentally different model than ours. We cite it in the revised manuscript to provide more background:

Note that the apparent fault strength is derived based on the assumption of the thermal weakening with exponential slip-weakening as observed in laboratory experiments⁶² and the complexity of fault roughness⁷⁶ is not considered.

14. line 181: 'discussions' -> 'discussion'

A. Revised.

15. line 221: 'and' to 'whereas'

A. Revised.

16. lines 222-242: this paragraph is based on V_c but there is no physical meaning of V_c yet. The cut-off velocity in the specific RSF used in this paper is not a standard format. Why not use the standard RSF format?

A. Please refer to our response to question 2.

REVIEWER COMMENTS

Reviewer #1 (Remarks to the Author):

My comments have been almost fully addressed. I congratulate the authors for their work.

There is one minor comment that in my opinion has been dismissed a bit too quickly, and I would recommend the authors to reconsider it. In my previous comment (#4), I pointed out that some statements in the manuscript, while correct, neglect the fact that the fault stress state might also play a role (ie in the energy release rate). The authors then argued in their response that in cycle simulations, the stress state would be governed by the friction behavior (which is true), but that they cannot run cycle simulations for computational reasons. I agree with the authors that cycle simulations are beyond the scope of this work. However, since these are not cycle simulations, the initial stress is an important control parameter, and I believe that, for the sake of transparency, it would be appropriate to at least clarify this point in the manuscript and avoid those ambiguous statements.

Reviewer #2 (Remarks to the Author):

I am satisfied with the revised manuscript. For me, the manuscript is acceptable for publication.

Reviewer #3 (Remarks to the Author):

The authors have addressed some of my previous comments. I only have a few questions in the revised manuscript.

why a new author added in the revised paper?

The physics of the numerical model presented in this paper mostly depends on 'finite-width' rupture on nature faults. However, there is no explanation for the generality of such an assumption based on global observations. I would say most recent earthquakes in either crustal continents e.g. multi-fault 2021 M7.4 Maduo, 2019 M6.4+M7.2 Ridgecrest, or subduction tectonics e.g. 2011 M9.0 Tohoku-Oki, 2020 M7.8 Simeonof and 2021 M8.2 Alaska earthquakes, as well as SSEs in New Zealand, Mexican subduction, eastern Alaska and southwest Japan, violate such an assumption. I am wondering how should these earthquakes compile in this theory. Do the author have a statistic plot of W and L for world-wide SSEs and/or earthquakes?

if "The heterogeneity of shear stress has been widely reported in cycle simulations even when prescribed friction properties are uniform (e.g., Dal Zilio et al., 2020)" then what is the novelty in this model? I am not convinced with this reply. In addition, the authors say "the constrained effective normal stress is 1-10 kPa on the deep San Andreas fault and $\sigma > 1$ MPa in the Izu-Bonin Trench, which is attributed to differences of fluid pressure." But the appearance of SSEs in Cascadia is also contributed to pore fluid pressure in most numerical simulations. And thus why Cascadia's SSEs are explained by shear heterogeneity? Or can the variation in effective normal stress arise from different overloading?

Comments from reviewer #1:

1. There is one minor comment that in my opinion has been dismissed a bit too quickly, and I would recommend the authors to reconsider it. In my previous comment (#4), I pointed out that some statements in the manuscript, while correct, neglect the fact that the fault stress state might also play a role (ie in the energy release rate). The authors then argued in their response that in cycle simulations, the stress state would be governed by the friction behavior (which is true), but that they cannot run cycle simulations for computational reasons. I agree with the authors that cycle simulations are beyond the scope of this work. However, since these are not cycle simulations, the initial stress is an important control parameter, and I believe that, for the sake of transparency, it would be appropriate to at least clarify this point in the manuscript and avoid those ambiguous statements.

A. We revised the text as:

which depends on the frictional behaviour **and the resulting stress states** on the fault

G_c/G_0 is controlled by the specific frictional behaviour **and the resulting stress states** of the fault⁵⁷.

Comments from reviewer #3:

1. why a new author added in the revised paper?

A. The revised manuscript has a new author because he contributed multiple ideas to the previous versions as acknowledged in the previous versions and both authors had an equal role in developing the revisions.

2. The physics of the numerical model presented in this paper mostly depends on 'finite-width' rupture on nature faults. However, there is no explanation for the generality of such an assumption based on global observations. I would say most recent earthquakes in either crustal continents e.g. multi-fault 2021 M7.4 Maduo, 2019 M6.4+M7.2 Ridgecrest, or subduction tectonics e.g. 2011 M9.0 Tohoku-Oki, 2020 M7.8 Simeonof and 2021 M8.2 Alaska earthquakes, as well as SSEs in New Zealand, Mexican subduction, eastern Alaska and southwest Japan, violate such an assumption. I am wondering how should these earthquakes compile in this theory. Do the author have a statistic plot of W and L for world-wide SSEs and/or earthquakes?

A. Previous compilations of global earthquakes and SSEs have already shown that most of the largest earthquakes have rupture lengths larger than their widths (e.g., Figure 2a in Weng and Ampuero, 2019) and a large fraction of global SSEs have $L/W > 2$ (e.g., Figure 1 in Gao et al., 2012). Specifically, most of the compiled Cascadia SSEs have aspect ratios larger than 2 (e.g., Michel et al., 2019). To acknowledge this discussion, we revised the text as:

Here, we show that the rupture propagation of SSEs and earthquakes on long faults, **with elongated ruptures as widely observed**^{14,46,53}, can be predicted by the same theoretical equation of motion of the

rupture tip and the debated scaling behaviours of SSEs can be attributed to different types of fault heterogeneities.

References:

The Dynamics of Elongated Earthquake Ruptures. H. Weng and J. P. Ampuero Journal of Geophysical Research: Solid Earth 2019

Similar scaling laws for earthquakes and Cascadia slow-slip events. S. Michel, A. Gualandi and J.-P. Avouac Nature 2019 Vol. 574 Issue 7779 Pages 522-526

Scaling relationships of source parameters for slow slip events. H. Gao, D. A. Schmidt and R. J. Weldon Bulletin of the Seismological Society of America 2012 Vol. 102 Issue 1 Pages 352-360

3. if “The heterogeneity of shear stress has been widely reported in cycle simulations even when prescribed friction properties are uniform (e.g., Dal Zilio et al., 2020)” then what is the novelty in this model? I am not convinced with this reply.

A. The novelty of this model is not that “the heterogeneity of shear stress can produce a cubic moment-duration scaling” but that “heterogeneity of effective normal stress produces a linear scaling, which reconciles the debated scaling relations” (text quoted from our abstract). The latter is a novel explanation provided by our newly-developed theory. This result reconciles the two different scaling relations that have been proposed for slow earthquakes and that are currently under debate. We believe this novelty has been sufficiently highlighted in the paper: in the abstract (see text quoted above), introduction (“the debated scaling behaviours of SSEs can be attributed to different types of fault heterogeneities”) and closing paragraph (“This fundamental model ... reconciles the debated scaling relations”).

In addition, the authors say “the constrained effective normal stress is 1-10 kPa on the deep San Andreas fault and $\sigma > 1$ MPa in the Izu-Bonin Trench, which is attributed to differences of fluid pressure.” But the appearance of SSEs in Cascadia is also contributed to pore fluid pressure in most numerical simulations. And thus why Cascadia’s SSEs are explained by shear heterogeneity? Or can the variation in effective normal stress arise from different overloading?

A. In our model, the appearance of SSEs in Cascadia is also attributed to pore fluid pressure. What we demonstrate in this model is that the heterogeneity of shear stress can explain the cubic moment-duration scaling of SSEs as observed in Cascadia, while the heterogeneity of effective normal stress can explain the observed linear scaling based on a global compilation of SSEs from different fault environments, because effective normal stresses can vary significantly among different environments.